# Analysis of the nischarin expression across human tumor types reveals its context-dependent role and a potential as a target for drug repurposing in oncology

**Marija Ostojić, Ana Đurić [ID], Kristina Živić [ID], Jelena Grahovac [ID]** *

Department of Experimental Oncology, Institute for Oncology and Radiology of Serbia, Belgrade, Serbia

* jelena.grahovac@ncrc.ac.rs

**Data Availability Statement:** The datasets presented and analyzed in the current study are publicly available in online repositories: • The

## Abstract

Nischarin was reported to be a tumor suppressor that plays a critical role in breast cancer initiation and progression, and a positive prognostic marker in breast, ovarian and lung cancers. Our group has found that nischarin had positive prognostic value in female melanoma patients, but negative in males. This opened up a question whether nischarin has tumor type-specific and sex-dependent roles in cancer progression. In this study, we systematically examined in the public databases the prognostic value of nischarin in solid tumors, regulation of its expression and associated signaling pathways. We also tested the effects of a nischarin agonist rilmenidine on cancer cell viability *in vitro*. Nischarin expression was decreased in tumors compared to the respective healthy tissues, most commonly due to the deletions of the nischarin gene and promoter methylation. Unlike in healthy tissues where it was located in the cytoplasm and at the membrane, in tumor tissues nischarin could also be observed in the nuclei, implying that nuclear translocation may also account for its cancer-specific role. Surprisingly, in several cancer types high nischarin expression was a negative prognostic marker. Gene set enrichment analysis showed that in tumors in which high nischarin expression was a negative prognostic marker, signaling pathways that regulate stemness were enriched. In concordance with the findings that nischarin expression was negatively associated with pathways that control cancer growth and progression, nischarin agonist rilmenidine decreased the viability of cancer cells *in vitro*. Taken together, our study lays a ground for functional studies of nischarin in a context-dependent manner and, given that nischarin has several clinically approved agonists, provides rationale for their repurposing, at least in tumors in which nischarin is predicted to be a positive prognostic marker.

## Introduction

Nischarin was first identified in the year 2000 as a novel protein interacting with the α5 integrin subunit involved in the control of cell migration [1]. Soon it was recognized that it was the same protein as the imidazoline receptor antisera-selected protein (IRAS), at the time studied

Human Protein Atlas: https://www.proteinatlas.org/ENSG00000010322-NISCH/tissue https://www.proteinatlas.org/ENSG00000010322-NISCH/pathology • Gene Expression Profiling Interactive Analysis, version2 (GEPIA2): http://gepia2.cancer-pku.cn • UALCAN portal: http://ualcan.path.uab.edu/ • UniProt: https://www.uniprot.org/ https://www.uniprot.org/uniprotkb/Q9Y2I1 • Ensembl: https://www.ensembl.org/ • cBioPortal platform: http://www.cbioportal.org/ • Mexpress: https://mexpress.be • Broad Institute website: https://gdac.broadinstitute.org/ • TCGA Research Network: https://www.cancer.gov/tcga.

**Funding:** This research was supported by the Science Fund of the Republic of Serbia, PROMIS Grant No. 6056979, REPANCAN to JG; by the Ministry of Education, Science and Technological Development of the Republic of Serbia Agreement No. 451-03-68/2022-14/200043 to all authors; and the European Union's Horizon 2020 research and innovation programme under the Marie Skłodowska-Curie grant agreement No. 891135 to JG.

**Competing interests:** The authors have declared that no competing interests exist.

as a novel target in drug discovery [2]. Implications of involvement in regulation of cell movement and its potential as a druggable receptor made it an interesting target in cancer research. Over the past 20 years nischarin (NISCH) role has been studied mostly in the breast, ovarian [3] and lung cancer [4, 5]. Seminal work on elucidating NISCH role in breast cancer initiation and progression has been done by the Alahari group [6]. They have shown that NISCH was involved in breast epithelial cell migration and invasion through regulation of the Rac driven signaling and interaction with multiple proteins involved in formation of focal adhesions and invadopodia [7–11]. It was suggested that NISCH functions as a scaffolding protein integrating extracellular to intracellular signaling. Alahari group also developed a NISCH mutant mouse and showed that NISCH can interact with and activate AMPK thus having a role in the regulation of cell metabolism [12]. When crossed with the MMTV-PyMT mice (a mouse strain in which the oncogenic polyoma virus middle T antigen is driven by the mouse mammary tumor virus promoter), NISCH mutant mice had increased breast tumor growth and metastasis [13]. They further developed a NISCH exon 5 and 6 knock-out mouse and showed that NISCH KO mouse embryonic fibroblasts had increased migration, but lower oxygen production rates and lower ATP production [14]. This confirmed that NISCH was involved in several biological processes important for cancer progression. Of importance, a difference in metabolic phenotype of male and female NISCH KO mice was observed, in body fat distribution, insulin resistance and glucose tolerance [15].

Nischarin gene is on the 3p21.1 chromosome, location marked as a putative tumor suppressor cluster [16]. Cancer-specific methylation of the *NISCH* gene was found in the breast [7], ovarian [3], lung [4], head and neck, and gastric cancers [17] and *NISCH* loss of heterozygosity was reported in breast [7] and ovarian cancer [3]. *NISCH* mRNA expression was shown to be downregulated in breast [18] and ovarian [3] cancer tissue compared to the adjacent healthy tissue and the lack of expression has been proposed as a marker of cancer aggressiveness [6]. Driven by these findings, our group examined NISCH mRNA and protein expression in melanoma and was surprised to find that although NISCH was downregulated in melanoma tissue compared to the uninvolved skin, high *NISCH* expression associated with better prognosis only in female patients and was associated with the worse outcome in males [19]. In the gene set enrichment analysis *NISCH* associated with both overlapping and distinct signaling pathways in female and male melanoma patients. This prompted us to question the universality of the tumor suppressor role of NISCH in cancer. The aim of this study was to comparatively examine the prognostic value of *NISCH* in solid tumors, regulation of its expression and associated signaling pathways with special emphasis on the possible differences between male and female cancer patients. Ultimately, we tested the effects of the nischarin agonist rilmenidine on the viability of cancer cells *in vitro*.

## Materials and methods

### Nischarin expression in healthy and tumor tissues

Nischarin endogenous expression was assessed using an interactive web resource The Human Protein Atlas (HPA) [20, 21]. The HPA RNA-seq tissue data is reported as nTPM (normalized protein-coding transcripts per million), corresponding to mean values of the different individual samples from each tissue. For HPA protein expression overview, NISCH protein levels were detected in 45 tissues using the HPA023189 antibody. Protein levels are reported as not detected, low, medium, or high, based on the combination of the staining intensity and fraction of stained cells.

To determine how *NISCH* levels change in the tumor compared to the normal tissues, Gene Expression Profiling Interactive Analysis, version2 (GEPIA2) website [22, 23] was used. This

website combines RNA sequencing expression data of 9,736 tumors and 8,587 normal samples from the The Cancer Genome Atlas (TCGA) and the Genotype-Tissue Expression (GTEx) projects, using a standard processing pipeline. We visualized the different expression of *NISCH* between tumor and corresponding normal tissues using GEPIA2 default settings (differential method: limma, q-value cutoff = 0.01, log2FC (fold change) cutoff = 1, and "Match TCGA normal and GTEx data) and statistical significance was indicated on the output figure, which was downloaded and used in the manuscript.

Additionally, gene expression profiles with patients' information and survival data in 21 types of TCGA tumor samples were downloaded from the HPA [24, 25] in the format of Fragments Per Kilobase per Million (FPKM). Student's t-test was used to determine the difference in *NISCH* mRNA expression between female and male patients across different tumors, and a p-value < 0.05 represented the significant score.

## Nischarin protein expression in tumors

UALCAN portal [26, 27] was used to examine NISCH protein expression levels in primary tumors and normal tissue samples from The National Cancer Institute's Clinical Proteomic Tumor Analysis Consortium (CPTAC) dataset [28]. Integration and analysis of these data has been reported [29, 30]. Shortly, mass-spectrometry-based proteomic data from the CPTAC are presented as Z-values representing standard deviations from the median across samples for the given cancer type. Log2 Spectral count ratio values from CPTAC were first normalized within each sample profile, then normalized across samples. Data form the UALCAN website represent differential expression between comparison groups assessed using *t*-tests on log-transformed values, with two-sided p values. False discovery rates (FDRs) were estimated using the method of Storey and Tibshirani [30, 31].

HPA database [24] was used to obtain the NISCH protein expression levels in patients' tumor tissues stained with HPA023189 antibody, as well as the immunohistochemical staining images of the NISCH protein in tumor and normal tissues to determine its localization.

## Nischarin transcripts and isoforms

The GEPIA2 website was used to obtain the expression distribution and isoform usage distribution of all transcripts of *NISCH* gene in tumor and normal samples. Results of this analysis were paired with the information about isoforms' protein structure and transcript summary from the UniProt [32] and Ensembl websites [33, 34], respectively to detect protein coding transcripts and determine their usage distribution across all TCGA cancers and paired normal tissues. To visualize and determine the statistical significance of the difference in expression of *NISCH* transcripts between tumor and corresponding normal tissues we used GEPIA2 website default settings (differential method: limma, q-value cutoff = 0.01, log2FC (fold change) cutoff = 1, and "Match TCGA normal and GTEx data).

## Survival analysis

Patient overall survival analysis was sourced from the publicly available database the Human Protein Atlas (HPA). To obtain better resolution graphs as well as to determine hazard ratio (HR) values (logrank) with 95% confidence interval (CI) of ratio, primary tumor data for each cancer subtype (previously downloaded from the HPA website) was analyzed in GraphPad Prism using the HPA standard settings. Namely, the best cut-off value for *NISCH* expression, which refers to the *NISCH* expression value that yields maximal difference with regard to survival between the high and low *NISCH* expression groups at the lowest log-rank p-value, was determined for each cancer type and used to generate Kaplan-Meier plot. Samples for each

tumor type as well as individual sex subgroups were analyzed using log-rank (Mantel-Cox) test and p-value < 0.05 represented the significant score.

Additionally, we evaluated the outcome significance of *NISCH* expression across TCGA cancers optionally adjusted by clinical factors using the web resource TIMER2.0 [35, 36]. Gene_Surv (Gene_Outcome) module, which uses Cox proportional hazard model, was set to determine the influence of *NISCH* gene expression on survival time adjusted for patients' age, gender, and cancer stage. Output results were downloaded from the website.

## Genetic alterations

Somatic copy-number alteration (CNA) information together with complete genetic and mutation status for *NISCH* gene were analyzed using cBioPortal platform [37, 38]. NISCH copy number changes and mutation details from the TCGA PanCancer Atlas datasets for cancers in which *NISCH* showed to be a significant prognostic marker in the survival analysis were retrieved and plotted using GraphPad Prism. SKCM did not have available information for CNA in primary tumor samples and was therefore excluded from this analysis. To test whether genetic alterations have the effect on *NISCH* mRNA expression, t-test, one-way analysis of variance (ANOVA) and Dunnett's multiple comparisons test were used depending on how many groups of data were analyzed for each tumor type.

## Nischarin promoter methylation status

Preprocessed DNA methylation data for 14 cancers from the TCGA dataset (BLCA, COAD, HNSC, KICH, KIRC, KIRP, LIHC, LUAD, OV, PAAD, PRAD, SKCM, TGCT, and UCEC) were downloaded from Mexpress [39]. Plotted beta values were calculated as means of all beta values of the CpG probes located up to 1500bp upstream of *NISCH* transcription start site (TSS1500). Unpaired t-test was used to determine the difference between NISCH high and NISCH low group for each tumor type individually, and a p-value < 0.05 represented the significant score.

## Gene Set Enrichment Analysis (GSEA)

BLCA, COAD, GBM, HNSC, KICH, KIRC, KIRP, LIHC, LUAD, OV, PAAD, PRAD, SKCM, TGCT, THCA, and UCEC tumor samples' information were downloaded from the Broad Institute website [40]. HiSeq level 3 data was downloaded in the format of RSEM normalized counts for genes, and only samples used in the aforementioned survival analysis were analyzed with GSEA software using the Hallmark and KEGG gene sets to find gene expression signatures associated with *NISCH*. NISCH high and NISCH low phenotypes were defined according to the *NISCH* mRNA levels used as a cut off in the survival analysis for each cancer type. GBM, SKCM and THCA samples were previously divided into subgroups by sex. Additionally, Reactome gene set was used for SKCM and GBM. Metrics used for ranking genes were Signal2-Noise and 1000 permutations with permutation type set to phenotype. A significant gene set was defined as the one with a nominal p value < 0.05 and false discovery rate (FDR) < 0.25 [41, 42].

## Nischarin expression in cell lines

Nischarin mRNA levels in several cancer cell lines (MIA PaCa-2, PANC-1, HCT 116, HT-29, A-375, and Hs 294T) were assessed using the HPA website [43]. Obtained RNA expression data was visualized using Graph Pad Prism.

## Cell culture and treatment

Human pancreatic cancer cell lines MIA PaCa-2 and PANC-1 were cultured in Dulbecco's modified Eagle's medium (DMEM) with 4.5 g/L glucose (Sigma-Aldrich, D6429, USA), 10% FBS and antibiotics. Colon cancer (HCT 116 and HT-29) and melanoma (A-375 and Hs 294T) cell lines were maintained as a monolayer culture in the RPMI 1640 medium with 2g/l glucose (Sigma-Aldrich, R8755, Germany), 10% FBS and antibiotics. The effects of NISCH agonist rilmenidine on cell viability was determined by performing the MTT assay (Sigma-Aldrich, USA) with at least three biological repeats. Cells were seeded into 96-well plates at cell densities of 3000 cells/well (MIA PaCa-2 and A-375), 5000 cells/well (PANC-1, HCT 116 and Hs 294T) and 8000 cells/well (HT-29) and left overnight to adhere before the 72h continuous incubation with 0–200μM of rilmenidine (Rilmenidine hemifumarate salt, R134, Merck, Germany). The rilmenidine concentration of 0 corresponds to the control that is not treated with the investigated compound and with cell viability set as 100%. Absorbances were measured on Multiscan EX microplate reader (Thermo Labsystems, Finland) at a wavelength of 570 nm. The $IC_{50}$ values (concentration of the investigated compound that causes 50% decrease in the MTT reduction in treated cell population compared to a non-treated control) were determined from the dose response curves plotted using Graph Pad Prism.

Additionally, quantitative analysis of apoptotic cell death induced by rilmenidine was performed using an Annexin V-FITC apoptosis detection kit according to the manufacturer's instructions (BD Biosciences, USA). Briefly, $2 \times 10^5$ A-375 cells/well were seeded in 6-well plates and after overnight adhesion cells were treated with 0, 10, 50 or 100 μM rilmenidine. Following the 24 h and 48 h incubation times, cells were trypsinized, washed twice with ice-cold PBS, and resuspended in the Binding Buffer (10 mM HEPES/NaOH pH 7.4, 140 mM NaCl, 2.5 mM CaCl2). After 15 min of incubation with Annexin V-FITC and PI at room temperature in the dark, the cells were analyzed using a Becton–Dickinson FACSCalibur flow cytometer (San Jose, USA) and Cell Quest computer software. Results are analyzed by two-way ANOVA and Dunnett's multiple comparisons test, and represented as fold change in Annexin V-FITC fluorescence (early and late apoptosis) compared to control.

## Results

### Nischarin expression in healthy and cancer tissues

Nischarin mRNA and protein were expressed in all the examined HPA healthy human tissues (S1 Fig). While the majority of analyzed tissues showed medium NISCH expression, protein staining had a high score in cerebellum, adrenal gland, bronchus, rectum, gall bladder, heart muscle, and the skin.

Analysis of the *NISCH* mRNA expression across 31 human solid tumor types (abbreviations listed in Table 1) compared to the corresponding normal tissues showed that *NISCH* mRNA levels were decreased in most solid tumor types: ACC, BRCA, CESC, COAD, GBM, LUAD, LUSC, OV, PRAD, READ, SKCM, TGCT, THCA, UCEC, UCS (Fig 1A). Only in thymoma *NISCH* expression showed the opposite trend. Sex-related difference in *NISCH* expression was detected in two renal cancer subtypes–KIRC and KIRP (Fig 1B)–with higher *NISCH* levels in tumor samples obtained from females, while for other cancers expression in tissues from female and male patients were similar.

In the CPTAC database, data on protein expression was available on paired healthy and tumor tissue only for 10 tumor types: BRCA, COAD, GB, HNSC, KIRC, LIHC, LUAD, OV, PAAD and UCEC. NISCH protein levels were significantly lower in nine out of ten tumor types, all but pancreatic ductal adenocarcinoma in which it was lower, but not significantly

**Table 1. Abbreviations and full names of cancers examined in this study.**

| Abbreviation | Full name |
| --- | --- |
| ACC | Adrenocortical Cancer |
| BLCA | Bladder Urothelial carcinoma |
| BRCA | Breast invasive carcinoma |
| CESC | Cervical squamous cell carcinoma and endocervical adenocarcinoma |
| CHOL | Cholangiocarcinoma |
| COAD | Colon adenocarcinoma |
| CRC | Colorectal cancer |
| ESCA | Esophageal carcinoma |
| GBM | Glioblastoma multiforme |
| HNSC | Head and Neck squamous cell carcinoma |
| KICH | Kidney Chromophobe |
| KIRC | Kidney renal clear cell carcinoma |
| KIRP | Kidney renal papillary cell carcinoma |
| LGG | Brain Lower Grade Glioma |
| LIHC | Liver hepatocellular carcinoma |
| LUAD | Lung adenocarcinoma |
| LUSC | Lung squamous cell carcinoma |
| MESO | Mesothelioma |
| OV | Ovarian serous cystadenocarcinoma |
| PAAD | Pancreatic adenocarcinoma |
| PCPG | Pheochromocytoma & Paraganglioma |
| PRAD | Prostate adenocarcinoma |
| READ | Rectum adenocarcinoma |
| SARC | Sarcoma |
| SKCM | Skin Cutaneous Melanoma |
| STAD | Stomach adenocarcinoma |
| TGCT | Testicular Germ Cell Tumors |
| THCA | Thyroid carcinoma |
| THYM | Thymoma |
| UCEC | Uterine Corpus Endometrial Carcinoma |
| UCS | Uterine Carcinosarcoma |
| UVM | Uveal melanoma |

(Fig 2A). We inspected the protein expression in these 10 tumor types in the HPA samples stained with the HPA023198 antibody that recognizes protein products of all 4 reported *NISCH* transcripts (see below) and found that majority of tumor tissues had weak to moderate staining (S2 Fig). Breast cancer samples exhibited only cytoplasmic and membranous staining (Fig 2B), as previously reported [18], and the same staining pattern was observed in the endometrial cancer. Interestingly, the rest of the tumor types also had nuclear localization of NISCH, ranging from 10% of samples (glioma) to 50% of the stained samples (COAD, HNSC, LIHC, PAAD). Colon adenocarcinoma (Fig 2C) and hepatocellular carcinoma (Fig 2D) had moderate nuclear, cytoplasmic, and membranous staining, while in the lung adenocarcinoma the staining was weak and mostly cytoplasmic and membranous (Fig 2E). In renal cancer more than 50% of samples did not express NISCH at all. We have previously reported the nuclear localization of NISCH in melanoma [19]. While the number of tumor samples stained in the HPA was too low for the statistical analysis with an appropriate power, it would be

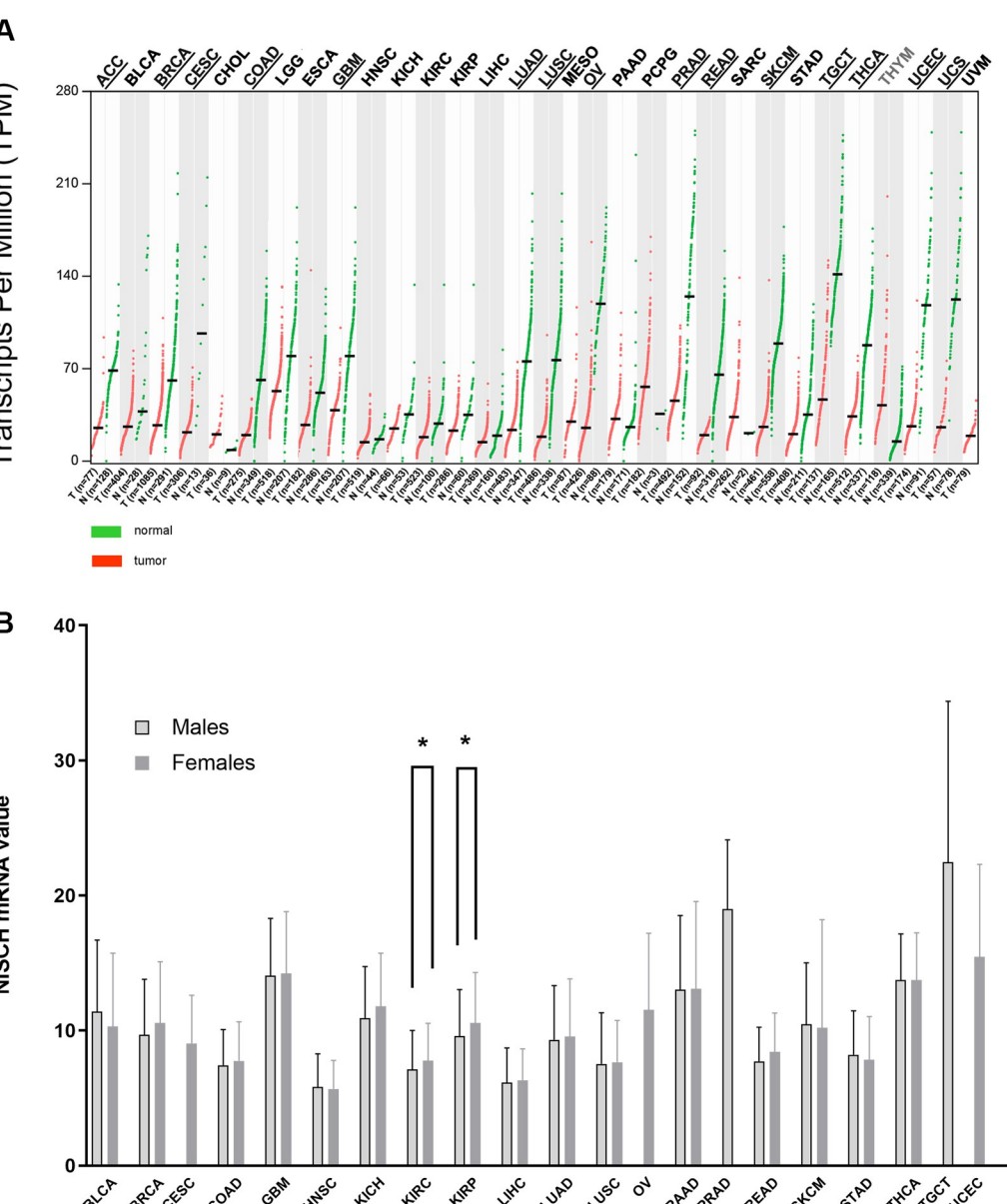

**Fig 1. The pan-cancer overview of the *NISCH* mRNA expression pattern.** (**A**) The gene expression profile in solid tumor samples and paired normal tissues from the TCGA and GTEx databases summarized by the GEPIA2. Underlined cancer type abbreviations indicate significantly higher *NISCH* expression in normal tissue compared to tumors, abbreviations in grey color indicate significantly higher expression in tumors (THYM). q-value cutoff = 0.01, log2FC (fold change) cutoff = 1. (**B**) Overview of the *NISCH* mRNA expression differences between female and male patients for each TCGA tumor type available on The Human Protein Atlas website. * p < 0.05.

interesting to examine (in each tumor type individually), whether the NISCH localized in the nucleus has a specific role and an impact on tumor progression.

## Nischarin isoforms in healthy and cancer tissues

NISCH has 4 isoforms produced by the alternative splicing [44]. Isoform 1 (Q9Y2I1-1, IRAS-1, IRAS-M), chosen as the canonical sequence, is the full-length protein (1504aa, 167kDa) that

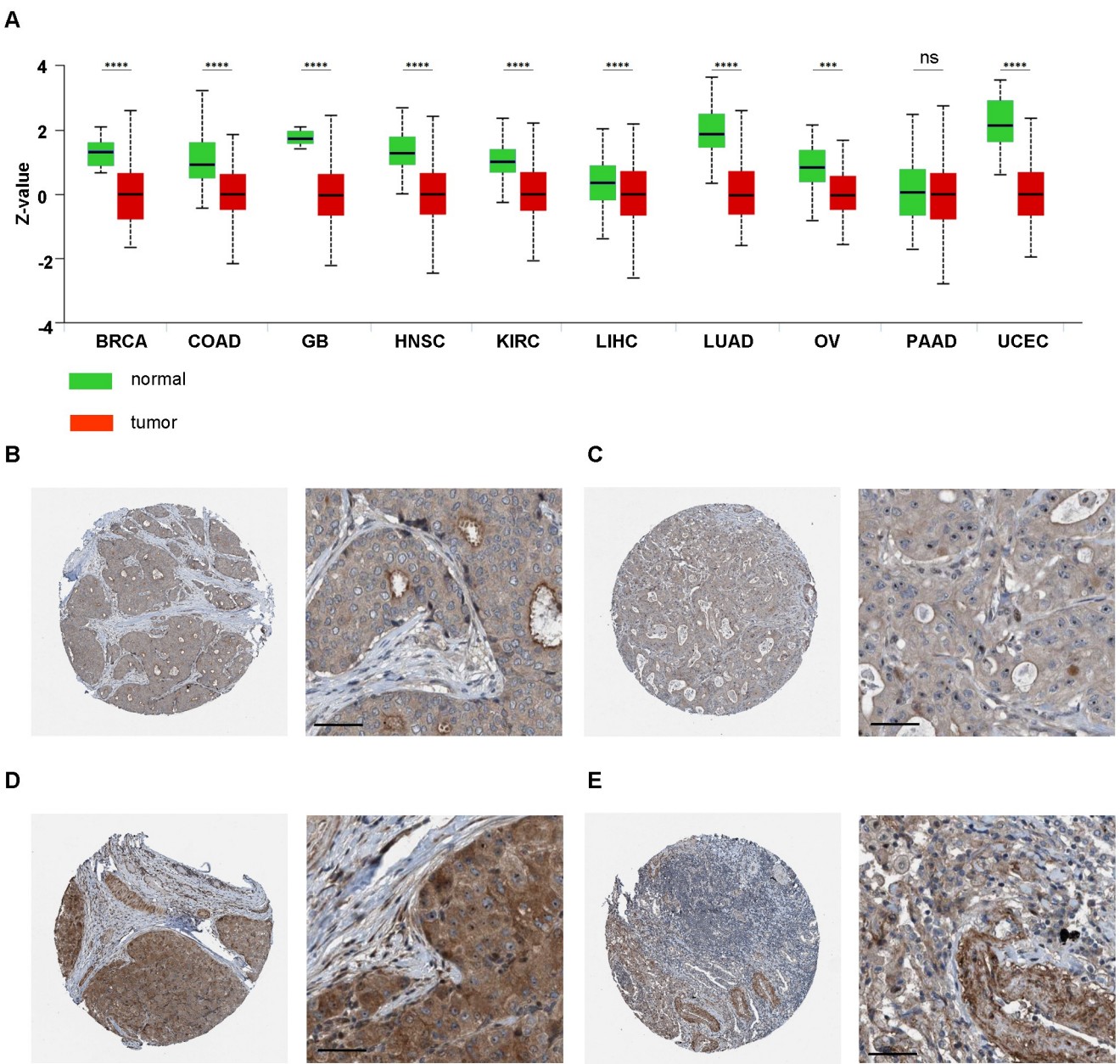

**Fig 2. Protein expression of NISCH in normal tissues and primary tumors.** (**A**) NISCH protein expression from the CPTAC database visualized using the UALCAN website. Z-values represent standard deviations from the median across samples for the given cancer type. Log2 Spectral count ratio values from CPTAC were first normalized within each sample profile, and then normalized across samples. ** $p < 0.01$, *** $p < 0.001$, **** $p < 0.0001$, by t-test. (**B**) BRCA, (**C**) COAD, (**D**) LIHC, and (**E**) LUAD NISCH immunohistochemistry staining images in human tumors from the HPA database. Inset with digital zoom, scale bar 50μm.

is highly expressed in neural and endocrine tissues [2, 45]. Isoform 2 (Q9Y2I1-2) is missing amino acids 1–511 which results in reduction of its length (993aa) [46]. Isoform 3 (Q9Y2I1-3, IRAS-L), with a modified sequence in 511-583aa and missing amino acids 584–1504 (583aa), is expressed dominantly in the brain as well as the isoform 4 (Q9Y2I1-4, IRAS-S) that has only 515aa since it is missing amino acids 516–1504 and differs from the canonical sequence in amino acids 512–515 [45]. As the isoforms 2, 3, and 4 are significantly shorter than isoform 1,

they lack sequence parts important for a full functioning NISCH protein. Namely, isoform 2 is missing an important part of N-terminus with PX domain, a domain of NISCH/IRAS that binds to phosphatidylinositol-3-phosphate in membranes [47], as well as leucine-rich region motifs important for protein-protein interactions. The PX domain together with the coiled-coil domain of NISCH is essential for its localization to endosomes, implying that NISCH isoforms 2, 3, and 4 may have different cellular localization. NISCH C-terminal domain, which is missing in the isoforms 3 and 4, interacts with IRS 1–4 and Rab14 [48, 49], and both the N- and C-terminus interact with Rac1 [9]. Considering their positions in the canonical sequence, other binding sites for partner proteins important for NISCH function are also compromised in some of the shorter isoforms (discussed in more detail in the section on mutation). Given that only isoform 1 is the full-length protein and all the other isoforms have impaired functions, we next investigated their expression patterns across tumor types and potential contributions to cancer progression.

To find the transcript summary for the NISCH splice variants in healthy and tumor tissues, Ensembl database [34] was used and 3 alternative protein-coding transcripts encoded by this gene were identified: ENST00000479054 and ENST00000345716 transcripts coding 1504aa protein matching UniProt's isoform 1 Q9Y2I1-1 and transcript ENST00000420808 coding 515aa protein matching UniProt's isoform 4 Q9Y2I1-4 form. There was also one computationally mapped potential NISCH isoform–C9J715, coded by the transcript ENST00000488380 [34], whose 583aa sequence has similarity of 99.3% with the isoform 3.

We analyzed the distribution of the 4 transcripts in solid tumors and matching healthy tissues using GEPIA2 website (Fig 3 and S3 Fig) and determined that the transcript ENST00000479054, coding isoform 1, was the dominant transcript in all cancer types as well as in normal tissues. Its expression significantly decreased in tumor compared to the healthy tissue in most of the cancer types, except for CHOL, PAAD, SARC and THYM where it increased, but only in THYM did the difference reach statistical significance. The isoform coded by ENST00000345716 transcript was expressed at low levels in healthy tissues and decreased or was even not detected in most of the tumor tissues that we examined. The second most present transcript in normal tissues was ENST00000488380 and its expression decreased in tumors. The ENST00000420808 transcript, that codes the isoform 4, was expressed in all the examined healthy and tumor tissues. Its levels mostly did not change, except in THYM, where they were increased in the tumor tissue, and OV, SKCM and TGCT, where the levels were decreased in tumor compared to the healthy tissue. Taken together, the most dominant NISCH transcript was the one coding for the full-length protein, and in the majority of the tumors most of the examined transcripts were decreased, implying that in cancer there is no NISCH isoform switching.

## Survival analysis

To examine whether the decreased *NISCH* mRNA expression in tumors had a prognostic value, we performed overall survival analysis of 20 tumor types (Table 2 and S1 Table). We found that *NISCH* was a favorable prognostic marker in seven of them: BLCA, HNSC, KIRP, LUAD, PAAD, TGCT, and UCEC (Fig 4A, S4 Fig), and unfavorable prognostic marker in another seven: CRC, KICH, KIRC, LIHC, OV, PRAD, and SKCM (Fig 4B, S4 Fig). For the analyzed breast cancer samples from the TCGA dataset, higher *NISCH* levels were associated with the better prognosis, but did not reach statistical significance (S4 Fig, p = 0.067). We were surprised that *NISCH* expression was an unfavorable prognostic marker in colorectal carcinoma (Fig 4B), as most of the CRC are adenocarcinomas. When we divided the samples into the COAD and READ subgroups, *NISCH* was an unfavorable prognostic marker in COAD, but

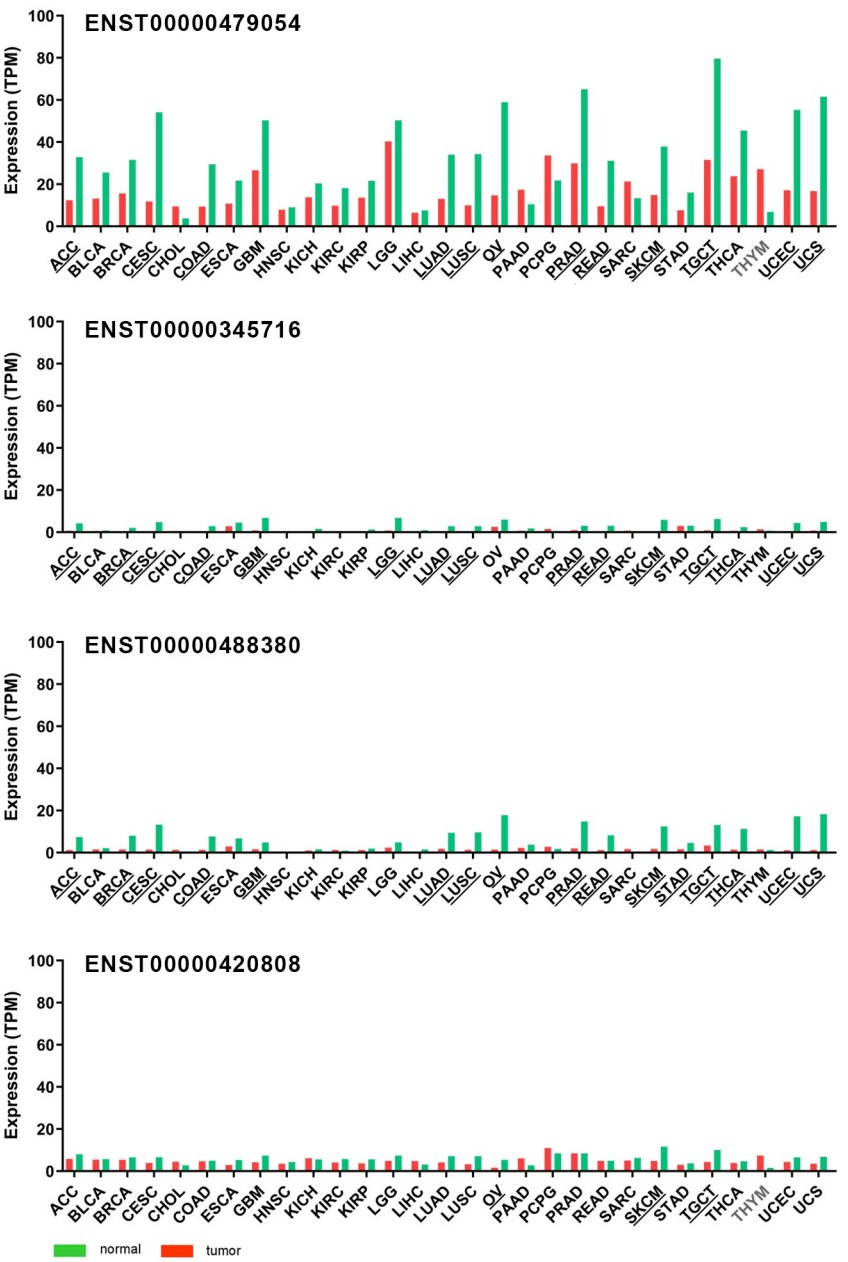

**Fig 3. *NISCH* transcript variants expression across available solid TCGA cancers and respective normal tissue.**
*NISCH* transcript variant ENST00000479054 is corresponding to the protein isoform 1, ENST00000345716 also corresponding to the protein isoform 1, ENST00000488380 predicted to be isoform 3, and ENST00000420808 coding isoform 4. Underlined cancer type abbreviations indicate significantly higher expression in the normal tissue compared to tumors, and abbreviations in gray color indicate significantly higher expression in tumors. q-value cutoff = 0.01, log2FC (fold change) cutoff = 1.

not in READ in which it had no significant prognostic value (Fig 4C). CRC pathogenesis depends on the anatomical location of the tumor, and it substantially differs between colon and rectum [50]. Our finding implies that the NISCH role may also differ between the two localizations. In addition, there is increasing evidence that COAD shows substantial differences in patient outcome depending on its origin: on the right or the left side of the colon [51].

**Table 2.** *NISCH* prognostic value in TCGA cancers, data from the human protein atlas.

| HPA cancer type | TCGA cancer | Survival analysis - best cut off | | | | | | | | |
|---|---|---|---|---|---|---|---|---|---|---|
| | | All primary samples | | | Females | | | Males | | |
| | | No. of samples | prognostic marker | P-val | No. of samples | prognostic marker | P-val | No. of samples | prognostic marker | P-val |
| Urothelial cancer | BLCA | 406 | **favorable** | 0.007 | 107 | **favorable** | 0.047 | 299 | **favorable** | 0.025 |
| Breast cancer | BRCA | 1075 | ns | 0.067 | 1063 | ns | 0.083 | 12 | ns | 0.270 |
| Cervical cancer | CESC | 291 | ns | 0.158 | 291 | ns | 0.158 | x | x | x |
| Colorectal cancer | | 597 | *unfavorable* | 0.002 | 275 | *unfavorable* | 0.007 | 322 | *unfavorable* | 0.024 |
| | COAD | 438 | *unfavorable* | <0.001 | 204 | *unfavorable* | <0.001 | 234 | *unfavorable* | 0.010 |
| | READ | 159 | ns | 0.270 | 71 | ns | 0.196 | 88 | ns | 0.456 |
| Glioma | GBM | 153 | ns | 0.148 | 54 | ns | 0.139 | 99 | ns | 0.126 |
| Head and neck cancer | HNSC | 499 | **favorable** | 0.008 | 133 | ns | 0.120 | 366 | **favorable** | 0.023 |
| Renal cancer | KICH | 64 | *unfavorable* | 0.031 | 26 | ns | 0.065 | 38 | ns | 0.110 |
| | KIRC | 528 | *unfavorable* | 0.009 | 184 | ns | 0.190 | 344 | *unfavorable* | 0.007 |
| | KIRP | 285 | **favorable** | 0.024 | 76 | favorable | 0.015 | 209 | **favorable** | 0.024 |
| Liver cancer | LIHC | 365 | *unfavorable* | 0.016 | 119 | *unfavorable* | 0.021 | 246 | *unfavorable* | 0.040 |
| Lung cancer | LUAD | 500 | **favorable** | <0.001 | 270 | favorable | 0.016 | 230 | **favorable** | 0.011 |
| | LUSC | 494 | ns | 0.243 | 128 | ns | 0.124 | 366 | ns | 0.491 |
| Ovarian cancer | OV | 373 | *unfavorable* | 0.008 | 373 | *unfavorable* | 0.008 | x | x | x |
| Pancreatic cancer | PAAD | 176 | **favorable** | <0.001 | 80 | favorable | <0.001 | 96 | **favorable** | 0.005 |
| Prostate cancer | PRAD | 494 | *unfavorable* | 0.014 | x | x | x | 494 | *unfavorable* | 0.014 |
| Melanoma | SKCM | 102 | *unfavorable* | 0.050 | 42 | favorable | 0.045 | 60 | *unfavorable* | 0.002 |
| Stomach cancer | STAD | 354 | ns | 0.119 | 125 | ns | 0.110 | 229 | ns | 0.300 |
| Testis cancer | TGCT | 134 | **favorable** | 0.029 | x | x | x | 134 | **favorable** | 0.029 |
| Thyroid cancer | THCA | 501 | ns | 0.147 | 366 | ns | 0.053 | 135 | ns | 0.116 |
| Endometrial cancer | UCEC | 541 | **favorable** | 0.004 | 541 | favorable | 0.004 | x | x | x |

Differences arise from the fact that, although colon is one organ, it develops from two distinct embryonic areas of the primitive gut that have localization-specific characteristics leading to two unique malignancies [52]. Therefore, we divided colon cancer samples into the left- and the right-sided groups, but there was no significant difference in *NISCH* expression or patients' survival (S5 Fig). Taken together, high level of *NISCH* expression was not a universal marker of good prognosis in cancer patients, not even within the same tumor group (such are adeno-carcinomas BRCA, COAD, PAAD, LUAD, etc.), as it was previously postulated.

To test whether *NISCH* had different prognostic values in female and male patients, tumor samples were divided into sex subgroups and the best cut off value for survival analysis was applied for each newly formed subset of samples (Table 2). As we previously reported [19], *NISCH* had the opposite prognostic value for female and male melanoma patients (Fig 4D). In some tumor types, like KIRC (unfavorable) and HNSC (favorable) the prognostic value stemmed from the male patient population that was represented 3 times more frequently but had the same trend in female patients (S6 Fig). In other tumor types, like THCA and GBM, the prognostic value, although it did not reach significance, had an opposite trend in female and male patients. Similar to the SKCM, female GBM patients with higher *NISCH* levels had better prognosis than males (Fig 4D). Glioblastoma and melanoma, derived from transformed glial cells and melanocytes, respectively, have a common neuroectodermal embryonic origin, and this may have accounted for a similar pattern. On the contrary, in the THCA the opposite

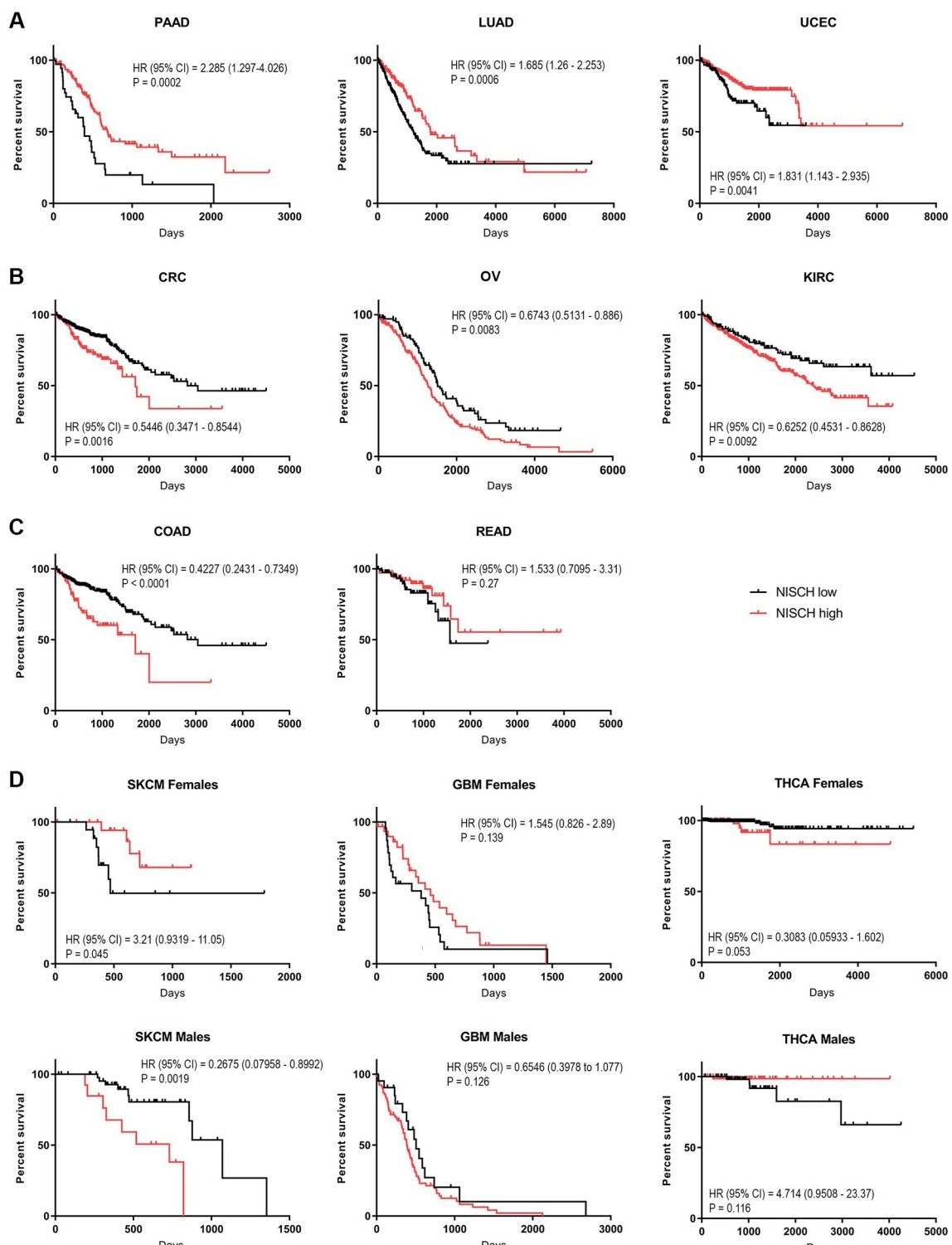

**Fig 4. *NISCH* prognostic value across different tumors.** Kaplan-Meier plots for: (**A**) selected tumors in which *NISCH* was a favorable prognostic marker, (**B**) selected tumors in which *NISCH* was an unfavorable prognostic marker, and (**C**) COAD and READ. (**D**) The effects of *NISCH* mRNA expression on overall survival of patients of the opposite sex in SKCM, GBM and THCA. HR = hazard ratio (logrank); CI = confidence interval of ratio; p = log-rank (Mantel-Cox) test p value.

pattern was present: males with higher *NISCH* expression and females with low *NISCH* expression had better prognosis (Fig 4D). Thyroid carcinoma is very heterogeneous, consisting of at least 5 histological types, with different clinical course [53, 54]. In the TCGA cohort that we examined, there were almost 3 times more samples of female patients, as female patients more often get diagnosed at an early stage [55]. Therefore, before drawing general conclusions, *NISCH* prognostic value in thyroid cancer should be examined in more detail by the histological type and stage, which was beyond the scope of this study. Nevertheless, the discrepancy in the prognostic value of *NISCH* in SKCM, GBM and THCA may indicate sex-related differences in NISCH signaling in these tumor types and is worthy of further investigation.

### Regulation of the NISCH gene expression

Since both NISCH mRNA and protein levels showed a decrease in most human tumor tissues compared to the respective healthy tissues, we examined the possible mechanisms of NISCH downregulation in tumor types in which levels of *NISCH* mRNA had a prognostic value. Loss of NISCH expression was previously reported as a consequence of a loss of heterozygosity and microdeletions of the *NISCH* gene in breast cancer [7] and *NISCH* promoter hyper-methylation in ovarian cancer [3]. Here, we examined the levels of *NISCH* promoter methylation and the presence of mutations and copy-number alterations (CNA) in the nischarin gene in cancers in which *NISCH* showed to be a significant prognostic marker. Mutations in the *NISCH* gene may influence *NISCH* mRNA transcription and stability but may also have an effect on the nischarin function as a tumor suppressor.

### Mutations

Mutations in the *NISCH* gene were present across the length of the gene, with no specific clustering (Fig 5A). They were present in most of the examined cancers, but at a very low frequency (Fig 5B, S7 Fig). The majority of mutations were missense and were the most frequent in UCEC (around 7%), SKCM (4%), BLCA (3.5%) and COAD (around 3%). Nevertheless, they had no significant impact on the *NISCH* mRNA expression level. Some of the mutations were predicted to be in the domains of NISCH that are specific binding sites for partner proteins important for NISCH localization and migration signaling. Mutations were detected in the PX domain (Fig 5A), a domain of NISCH/IRAS that binds to phosphatidylinositol-3-phosphate in membranes [47]. The PX domain together with the coiled-coil domain of NISCH is essential for its localization to endosomes, and although not all missense mutations may lead to an observable change in the function of the protein, the resulting NISCH protein may fail to function properly. This may be also true for other regions of NISCH/IRAS that interact with signaling molecules: positions 1–624aa that interact with PAK1 [56], 416–624aa region that interacts with LKB1 [10] and LIMK [11], positions 464-562aa that interact with the integrin α5 cytoplasmic tail [1], NISCH C-terminal domain that interacts with IRS 1–4 and Rab14 [48, 49], and both the N- and C-terminus that interact with Rac1 [9]. Throughout all these regions, mutations that are potentially harmful for NISCH function could be observed, although with a very low frequency for all analyzed tumors.

### Copy number alterations

Examination of the CNA in TCGA cancers (Fig 5C) showed that *NISCH* amplification was very rare in tumors, and gain of additional *NISCH* copy was not frequent. Most tumors had less than 10% of samples with additional *NISCH* copy, and the largest number of affected samples was detected in KIRP, around 25%. On the other hand, shallow deletion was the most common copy number alteration detected across all analyzed TCGA samples (Fig 5D, S8 Fig).

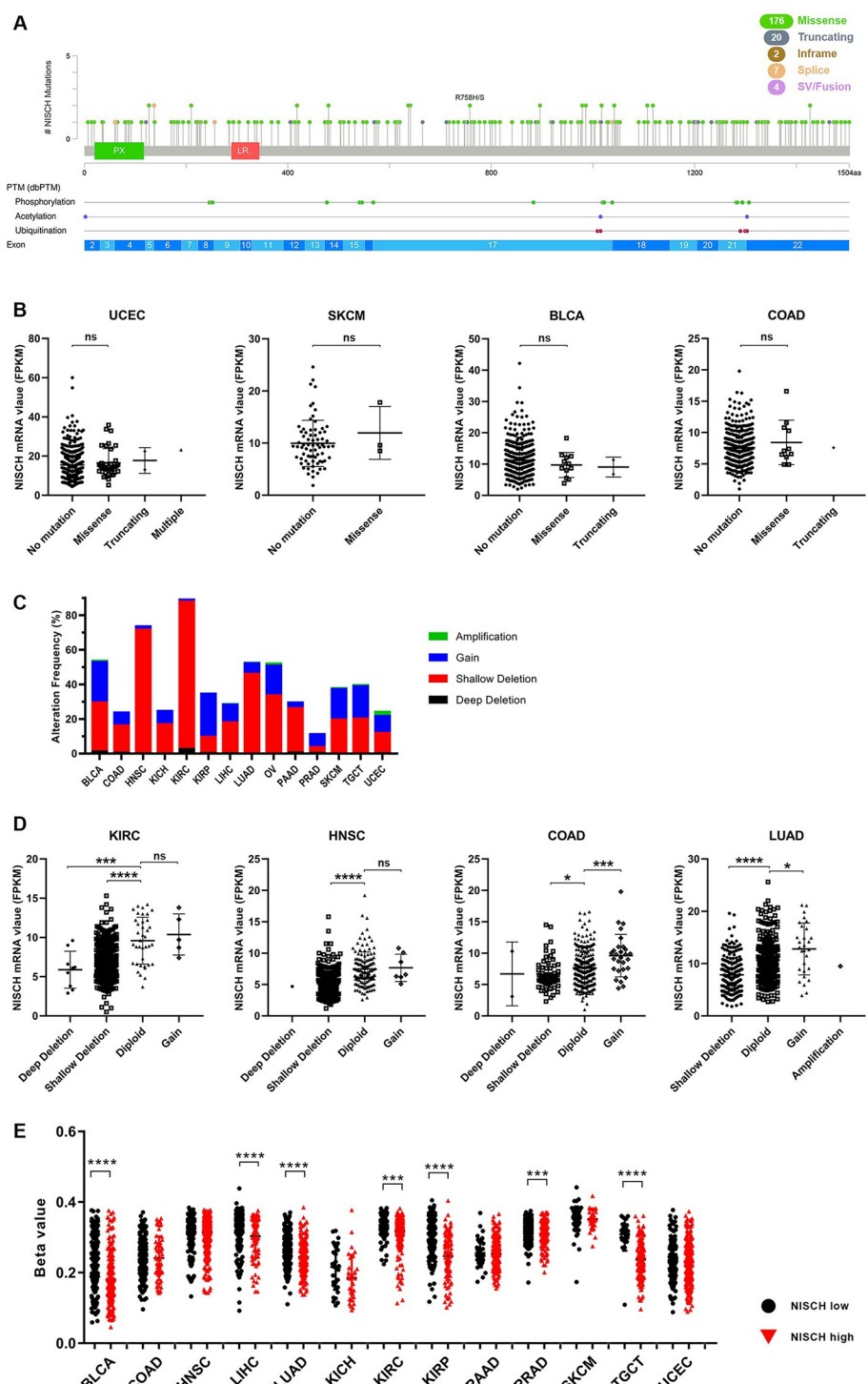

**Fig 5. Regulation of *NISCH* gene expression.** (**A**) Mutation diagram for the *NISCH* gene across TCGA tumors and (**B**) the effects of *NISCH* mutations on the levels of *NISCH* mRNA expression in UCEC, SKCM, BLCA and COAD. ns–not significant, by Dunnett's multiple comparisons test (No mutation = ctrl) (**C**) *NISCH* CNA frequency across TCGA cancer types and (**D**) the effect of CNA on *NISCH* mRNA expression in KIRC, HNSC, COAD and LUAD. ** $p < 0.01$, *** $p < 0.001$, **** $p < 0.0001$, Dunnett's multiple comparisons test (Diploid = ctrl) (**E**) *NISCH* promoter methylation status in cancers in which *NISCH* is a prognostic marker. ** $p < 0.01$, *** $p < 0.001$, **** $p < 0.0001$ by unpaired t-test (NISCH high vs NISCH low).

KIRC had the highest *NISCH* shallow deletion frequency, with this CNA detected in a striking 85% of profiled samples, followed by HNSC (72%), COAD (50%) and LUAD (46%). Most of the examined tumor types had a small number of samples affected by deep deletions, mostly around 1%, except KIRC with around 2.6% of samples with deep deletions present. These results lead to a conclusion that deep and shallow deletions of *NISCH*, defined as "possibly homozygous" and "possibly heterozygous" deletions, were frequent mechanisms for NISCH downregulation in cancer.

## Promoter methylation

Out of the 14 TCGA cancers analyzed for the presence of NISCH promoter methylation, seven types showed significant increase in methylation levels in tumor samples with lower NISCH expression compared to the samples with higher NISCH expression: BLCA, LIHC, LUAD, KIRC, KIRP, PRAD, and TGCT (Fig 5E). Ovarian cancer data set only had information for a single NISCH promoter probe as opposed to 10 probes for the other analyzed cancer types and was excluded from further analysis. Our results complement previous findings that NISCH promoter methylation is frequent in lung and kidney cancer [17, 57] and can affect NISCH expression. In genome-wide cancer methylome analysis [17], NISCH promoter methylation was not detected in pancreatic and colon cancer, and our results confirmed that methylation was not an important mechanism for NISCH downregulation in these types of cancer. Our results indicate that *NISCH* promoter methylation is frequent and can affect NISCH expression levels, but is not a universal mechanism for NISCH downregulation in cancer.

## Gene set enrichment analysis

Given that the incidence of mutations in the *NISCH* gene that could possibly account for a loss of a tumor suppressor function was very low, we performed gene set enrichment analysis (GSEA) to examine the observed disparity in its prognostic value in different cancers. To find gene expression signatures associated with *NISCH* expression we defined "NISCH high" and "NISCH low" phenotype according to the *NISCH* mRNA levels used as a cut off in survival analysis for each cancer type (Fig 6A, S2 and S3 Tables). The majority of signaling pathways that were commonly enriched in the "NISCH low" phenotype in all the examined tumor types were related to the increased metabolic activity: glycolysis, mTORC1 signaling, oxidative phosphorylation, gluconeogenesis, tricarboxylic acid (TCA) cycle, fatty acid metabolism, adipogenesis and different amino acids metabolism pathways (Fig 6A). These results imply that decreased *NISCH* expression in tumor tissues coincides with activation of pathways necessary for the increased tumor growth. Along these lines of increased production demands (under oncogenic and environmental stress during cancer progression), pathways involved in unfolded protein response, proteasome, DNA repair and reactive oxygen species pathway were all enriched in the "NISCH low" group regardless of the cancer type. These findings support the role of NISCH as a tumor suppressor in cancer.

Significantly enriched in the "NISCH high" phenotype in several cancer types (LIHC, OV, KIRC, LUAD, HNSC and TGCT) regardless of the *NISCH* prognostic role were inositol phosphate metabolism and phosphatidylinositol signaling (Fig 6A). These processes are involved in both cancer cell metabolism and regulation of the actin cytoskeletal rearrangement at the plasma membrane [58, 59], and were previously reported to be regulated by nischarin [6, 18].

Next, we looked to see whether there were differences in *NISCH* association with specific signaling pathways between cancers in which *NISCH* was an unfavorable and favorable prognostic marker (Fig 6B, S2 and S3 Tables). Wnt-beta catenin, Notch and Hedgehog signaling pathways were predominantly significantly positively associated with *NISCH* expression in the

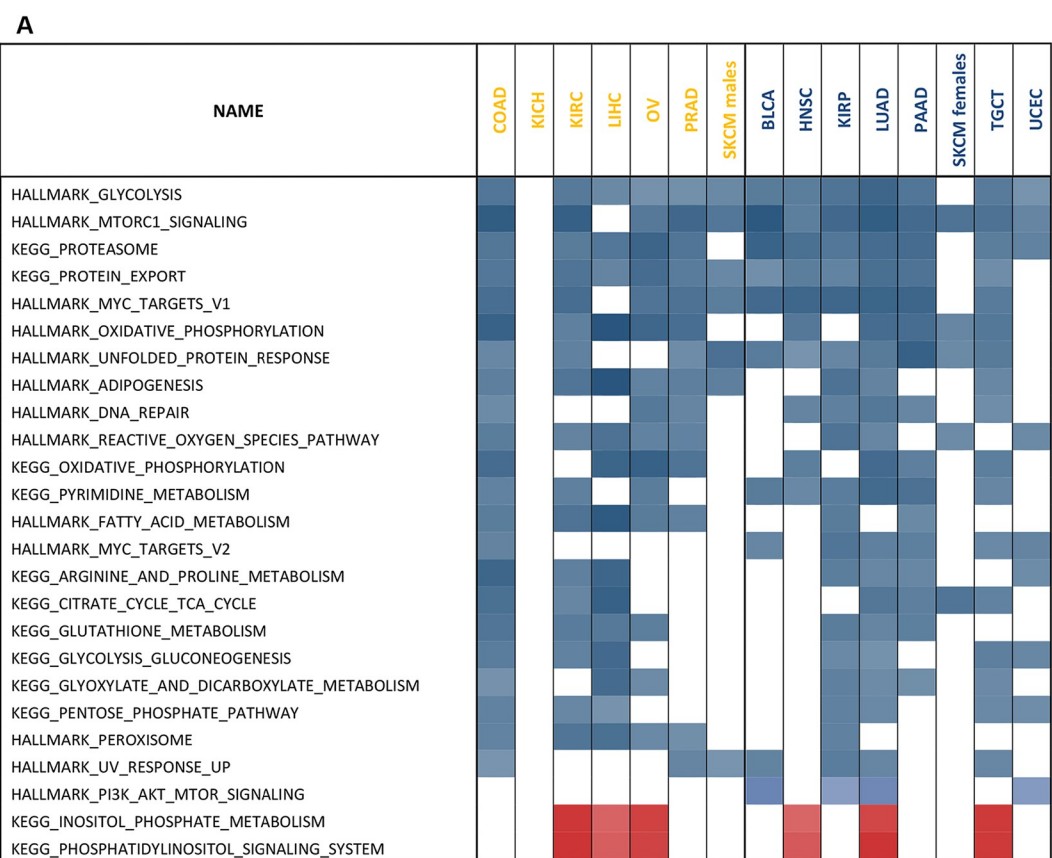

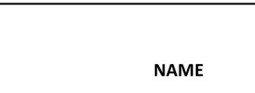

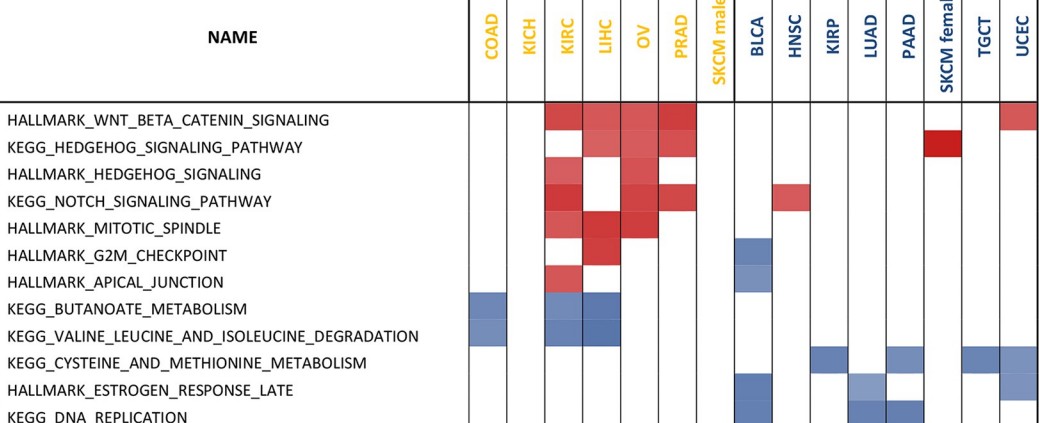

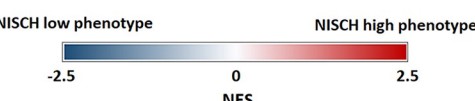

**Fig 6. Nischarin associated gene networks in solid tumors.** Results of the gene set enrichment analysis (GSEA) of primary tumors with (**A**) common association with *NISCH* expression and (**B**) opposite enrichment results depending on its prognostic value. A significant GSEA result is presented as normalized enrichment score (NES) with a nominal p value < 0.05 and false discovery rate (FDR) < 0.25. Cancer names in Yellow–unfavorable and Blue–favorable *NISCH* prognostic value.

group of cancers where *NISCH* was an unfavorable prognostic marker. Active Wnt signaling is a driver of cancer progression and chemoresistance in many types of cancer from different origin [60], including COAD, OV [61], LIHC [62], PRAD [63] and KIRC [64]. Together Wnt, Notch and Hedgehog signaling are involved in the maintenance of cancer stem cells [65] and provide resistance to various treatment modalities [66]. In LIHC, OV and KIRC, *NISCH* was also positively associated with the regulation of the mitotic spindle. Common thread for these cancer types was that nischarin was also present in the nucleus and with a relatively high number of gene copy alterations. These results suggest that NISCH localization in the nucleus and its consequences should be further investigated and that in cancers in which *NISCH* is an unfavorable prognostic marker there may be additional cellular processes in which NISCH is involved.

To examine the differences in *NISCH* prognostic value from the survival analysis, GBM, SKCM and THCA samples were divided into subgroups by sex. No common thread in analysis by sex was found by examining the pathways in the Hallmark and KEGG gene sets (S4 and S5 Tables). We further looked at the common pathways associated with *NISCH* by sex in the Reactome gene sets in melanoma and glioblastoma, as these two cancer types share the embryonic origin. Although there were pathways inversely associated with *NISCH* expression in male and female patients, again there were no common pathways by sex for these two types of cancer (S6 Table).

### *In vitro* effects of nischarin agonist rilmenidine on the viability of cancer cell lines

Given that *NISCH* expression was negatively associated with pathways that control cancer growth and progression, NISCH agonists may be of interest for repurposing in oncology. There are several FDA-approved nischarin agonists with good safety and tolerability profiles that are used for the treatment of hypertension: the first identified clonidine, and modified, the "second generation centrally acting drugs", moxonidine and rilmenidine with increased affinity and selectivity for nischarin [67, 68]. Here we tested rilmenidine, which previously showed the highest affinity for NISCH [68] To assess the differences in NISCH agonist effects between cancer types in which *NISCH* was a favorable and unfavorable prognostic marker, we chose two pancreatic cancer cell lines (MIA PaCa-2 and PANC-1) and two colon cancer cells lines (HCT 116 and HT-29). Additionally, we tested rilmenidine activity in two melanoma cell lines–A-375, derived from female melanoma patient, and Hs 294T, collected from a male patient–both of which carry BRAF mutation. To confirm that all the examined cell lines express NISCH, data from the HPA website was used to examine the levels of *NISCH* mRNA (Fig 7A). Next, we treated cancer cells with increasing concentrations of rilmenidine (0–200μM) for 72 h and measured cell viability by MTT assay (Fig 7B and 7C). Rilmenidine dose-dependently decreased viability in all the tested cell lines, where the least sensitive was the colon cell line HT-29 and the most sensitive the A-375 melanoma cell line. Reduction of the MTT dye in the viability assay may be a consequence of the cytotoxicity of the drug (induced cell death), the cytostatic effect (inhibition of the cell cycle) or the alteration of the cellular redox state. To determine the cause of the significant viability reduction of the A-375 cells in the MTT assay, we performed the Annexin-PI apoptosis assay after 24 and 48 h of treatment with up to 100μM of rilmenidine (Fig 7D). The results revealed that rilmenidine time- and dose-dependently induced apoptosis in tested melanoma cells (p < 0.0001 and p = 0.0005, respectively; interaction p value p = 0.0002 by two-way ANOVA). These results imply that activation of NISCH may have anti-cancer effects and is worthy of further investigation.

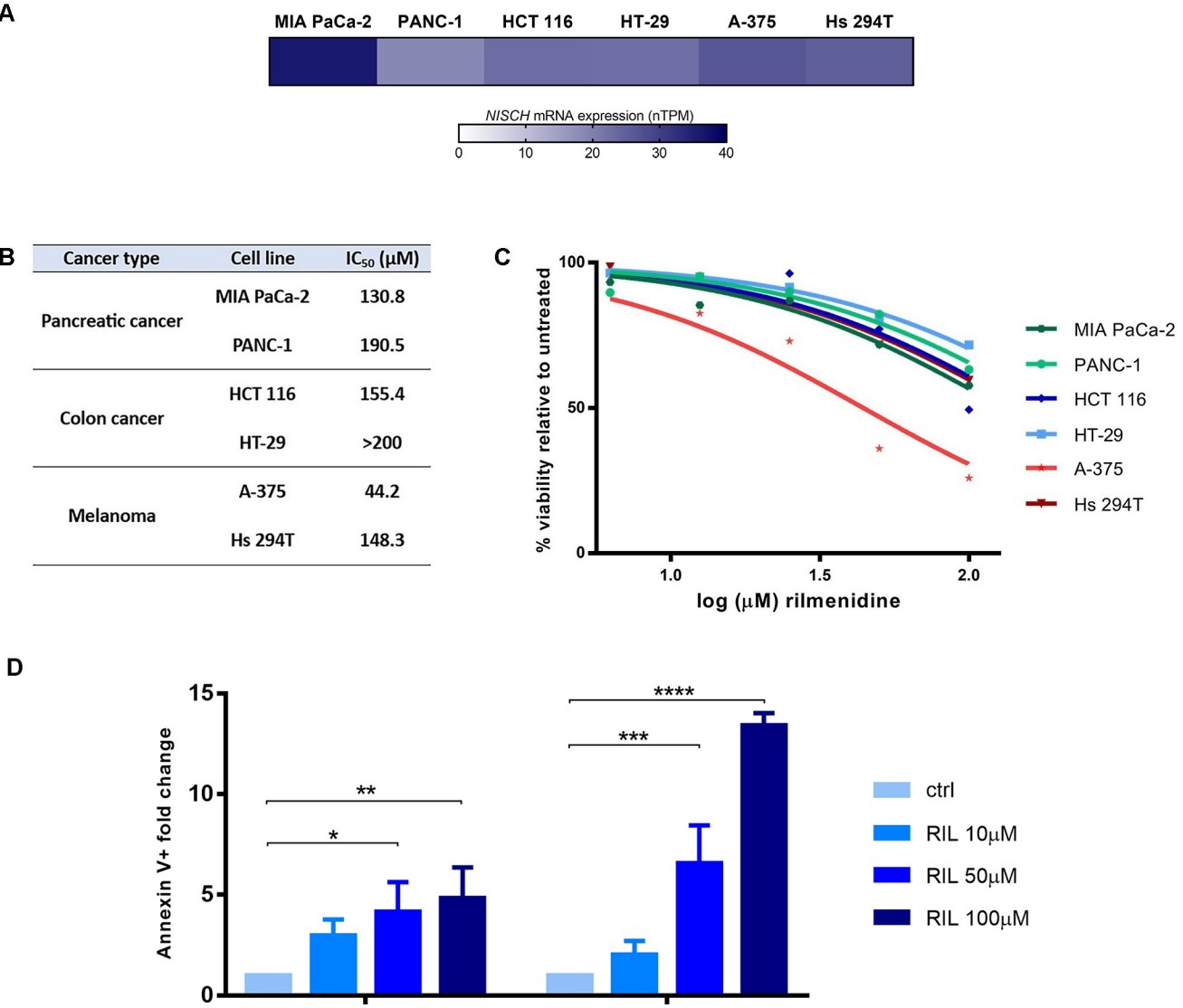

**Fig 7. *In vitro* effects of nischarin agonist rilmenidine on viability of cancer cell lines.** (**A**) Nischarin mRNA expression in pancreatic cancer, colon cancer and melanoma cell lines. Activity of rilmenidine presented as (**B**) $IC_{50}$ values and (**C**) percent of cancer cell viability after 72h of treatment by MTT test. (**D**) Fold change in number of Annexin-FITC positive A-375 cells (early and late apoptosis) after 24 h and 48 h of incubation with rilmenidine normalized to the untreated control. * $p < 0.05$, ** $p < 0.01$, *** $p < 0.001$, **** $p < 0.0001$, by Dunnett's multiple comparisons test.

## Discussion

Nischarin was so far known as a novel tumor suppressor gene whose downregulation promotes tumorigenesis [13], tumor progression [3, 6, 18], and poor survival in breast, ovarian and lung cancer patients [3, 5, 7, 18]. It was reported that exogenous expression of NISCH could suppress breast cancer cell survival and motility *in vitro* and growth *in vivo* [7, 18, 69, 70]. NISCH was found to be downregulated in breast and ovarian cancer tissues compared to the healthy counterparts and was found to be a marker of better prognosis [3, 6, 18]. Surprisingly, although NISCH expression was downregulated in melanoma tissue compared to the uninvolved skin, our group found that it was a favorable prognostic marker only in female

melanoma patients, but not in males [19]. Bearing in mind that most findings about NISCH role in cancer stemmed from the breast cancer research and that most of the examined patient data were from females, we aimed to perform a multidimensional pan-cancer analysis of nischarin in both sexes. We examined NISCH mRNA and protein expression, prognostic value, transcriptional regulation, as well as its potential role in cancer progression, by examining publicly available datasets, and considering sex-related differences.

NISCH was expressed at both the mRNA and protein level in all the examined healthy human tissues, and its expression was significantly decreased in most of the analyzed solid tumor types, except thymoma where it was increased. Thymoma is a rare type of tumor characterized by unique features in comparison to other tumors of epithelial origin. It rarely spreads beyond the thymus, has great histological heterogeneity, and is often associated with immune diseases [71], which makes it an outlier in pan-cancer analysis.

In addition to the dominantly cytoplasmic and membranous localization that was reported in healthy and breast cancer cells, our group reported that NISCH was also present in the cell nucleus in melanoma tissues [19]. This was a novel finding, as nuclear localization of NISCH in cancer cells has not been reported previously. Therefore, we examined the subcellular distribution of NISCH across tumors in the Human Protein Atlas and found that only in breast and endometrial cancer NISCH localization was restricted to the cytoplasmic and membranous. In the rest of the tumor types NISCH could also be found in the nucleus. This is a very interesting discovery because nuclear translocation of tumor suppressors is quite common in cancer to promote tumor development and inference of their localization is often implemented for diagnostic purposes [72]. The novel localization could also account for cancer-specific nischarin role and should be investigated in future functional studies. We analyzed the expression of NISCH by tumor type, and it is possible that analysis by molecular subtypes would reveal further intricacies of the NISCH role in cancer progression.

We next analyzed the distribution of the 4 transcripts coding NISCH isoforms and confirmed that isoform 1 coding the full-length protein was the dominant in both solid tumors and matching healthy tissues. The expression of transcript coding isoform 1 decreased in all cancer types except thymoma; and all other investigated transcripts had a similar pattern of decrease without the isoform switch between the healthy and the tumor tissue. Currently available antibodies for nischarin all bind to the region closer to the N-terminus, and therefore detect all the isoforms, but it would be interesting to investigate whether the nuclear localization is specific for a particular isoform.

Even though NISCH expression was decreased across most cancer types compared to the healthy tissues, higher expression in tumors was not a universally positive prognostic marker. Surprisingly, even within adenocarcinomas *NISCH* prognostic value was inconsistent: it was a positive prognostic marker in LUAD and PAAD, negative in COAD and PRAD, and had no prognostic value in READ and STAD. The difference between COAD and READ can be of exceptional importance since it was previously determined that the carcinogenic risk of the rectal mucosa to develop cancer is significantly higher than that of the colon mucosa [73]. NISCH was previously reported to be a positive prognostic marker in breast cancer [6], but in our examination of the TCGA dataset, it was not statistically significant. This may be due to the heterogeneity of breast cancer samples in the set. Okpechi et al [6] described that *NISCH* mRNA expression was lower in basal than in the luminal breast cancer, in ER negative compared to the ER positive and in PR negative compared to the PR positive tumor samples, but higher in HER negative tumors compared to the HER positive. They also reported that NISCH expression decreases with breast cancer stage and grade. This implies that NISCH prognostic role should further be examined in each tumor type by stage, grade, and molecular subtype. Surprisingly, our analysis implied that the higher *NISCH* mRNA expression was an

unfavorable prognostic marker in ovarian cancer patients, which was in contrast to the report by Li et al [3]. In that study, based on the immunohistochemical staining of NISCH in a tissue array consisting of 89 samples, it was found that increased NISCH protein levels were a positive prognostic marker.

Most of the cancers had lower NISCH levels compared to the healthy tissues, but only in some of them lower nischarin expression associated with outcome of the overall survival. To investigate the differences between tumors with the opposite prognostic value of *NISCH*, we examined the mutational status of *NISCH* gene and expression regulation mechanisms in cancer types in which the prognostic value, either negative or positive, was significant. Mutations in the *NISCH* gene were present across the length of the gene in most of the examined cancers in this study but had no significant impact on the *NISCH* mRNA expression level. A missense change L972P in the *NISCH* gene was previously shown to be an important factor in the development of otitis media and consequential conductive hearing loss in the mouse model [74]. Nevertheless, even though there was a possibility that mutations could affect some of the protein-protein interactions between NISCH and other signaling molecules, and disrupt its reported antitumor effects, they were present at a very low frequency and with no specific clustering to be considered as a cause for negative prognostic value of *NISCH* in a subset of tumors. In terms of the downregulation of NISCH expression, methylation of the *NISCH* promoter was an important factor in BLCA, LIHC, LUAD, KIRC, KIRP, PRAD, and TGCT, and shallow deletions were a common important mechanism for all the examined tumor types. This was not surprising, as studies have shown that allelic loss from several distinct regions on chromosome 3p, including 3p21–22 where *NISCH* gene is located, are the earliest and most frequent genomic abnormalities involved in a wide spectrum of epithelial cancers including lung, breast, kidney, head and neck, ovary, cervix, colon, pancreas, esophagus, and bladder [75]. Copy-number alteration frequency was the highest in KIRC and COAD, where NISCH had negative prognostic value, but similarly high CNA levels were noted in HNSC, in which *NISCH* was a favorable prognostic marker. Paradoxically, high levels of chromosomal instability can be both tumor suppressive, owing to the frequent generation of unviable karyotype, and tumor propagating, leading to high intratumoral heterogeneity, therapeutic resistance, and poor prognosis [76]. The complexity of this problem is the subject of many ongoing studies that attempt to better exploit the dynamic process of chromosomal instability in order to create new therapeutic opportunities in cancer.

It is worth mentioning that expression of NISCH may also be regulated by microRNAs. In HNSC, it was reported that miR-2355-5p decreased NISCH expression, leading to higher tumor cell proliferation [77], and knockdown of miR-23b and miR-27b was reported to upregulate nischarin and repress breast cancer growth [78]. However, OncomiR (an online resource that explores the miRNomes across TCGA cancers) [79] did not show significant correlations of *NISCH* mRNA with miRNAs in any of the cancers we examined (not shown).

It is possible that even without the mutations present, protein can have a tumor suppressive or promotive role that is context-dependent [80, 81]. We performed GSEA to look for the differences in associated gene networks in tumor types where *NISCH* had negative versus positive prognostic value. In line with the NISCH so far described biological roles [7–12], and regardless of the tumor type, decreased *NISCH* expression was associated with activation of metabolic pathways that allow increased tumor growth. In addition, in "NISCH low" phenotype, regardless of the prognostic role, pathways that characterize survival in hypoxic and nutrient deprived conditions like reactive oxygen species pathway, unfolded protein response [82], and DNA repair [83], were enriched. This is in line with our finding that nischarin levels are lower in most cancers compared to healthy tissues, since these are critical events for tumor progression. However, possible explanation for the differing prognostic role of NISCH in different

cancer types may be attributed exactly to the NISCH association with cancer metabolism. NISCH has an important role in the maintenance of the cellular metabolic homeostasis [15, 84, 85], and activated NISCH has a role in caloric restriction in tissues [86]. Cancers are heterogeneous in their metabolic dependencies and preferred energy sources, and this is influenced by the anatomical location and the microenvironment. Therefore, NISCH may have differing roles in cancers with different metabolic dependencies.

The pathways that were repeatedly associated with high *NISCH* expression in cancer types in which it had negative prognostic value were Wnt, Hedgehog and Notch signaling. It has been reported that NISCH may regulate some aspects of Wnt signaling [87] but the association of *NISCH* with Notch and Hedgehog signaling is a novel finding. Common characteristic for these three pathways is that they are cell-fate determining and their crosstalk is important in the maintenance of the cancer stem cell phenotype [65]. Whether there is a functional connection between nischarin and Wnt-Notch-Hedgehog is worthy of a functional study. The GSEA that we performed generated a plethora of cues that are worth investigating in functional assays in each tumor type separately, as NISCH complex association with distinct signaling pathways complicates elucidation of the full contribution of nischarin in the progression of different cancer types and NISCH role seems to be context dependent.

One of the limitations of our study is that our conclusions on the role of NISCH in cancer progression rely on the bulk tumor RNA sequencing and the gene set enrichment analysis. Namely, bulk sequencing data reflects complex phenotype composed from cancer cells and cancer stroma (cancer-associated fibroblasts, immune infiltrate, vasculature), and it is possible that depending on the tumor type, signal relies to a differing extent on the malignant versus non-malignant compartment. For example, up to 80% of the tumor mass in pancreatic cancer is derived from the desmoplastic reaction. However, this does not undermine the potential of NISCH agonists for repurposing, as they are systemic drugs and would have an effect on the whole tumor tissue. With the advancements of single cell sequencing the questions of NISCH cell origin in tumors may be resolved.

Ultimately, statistical difference does not always translate to the biological difference, as decrease of expression in a tissue with already low levels can have a greater biological impact than a decrease in high expressing tissues. For example, healthy rectal tissue has higher nischarin expression compared to the rest of the digestive tract, and decrease in NISCH had no prognostic value in rectal cancer.

Prompted by our previous study on melanoma, we set out to examine possible sex-related differences in NISCH prognostic role but have only found differences in survival by sex in two other cancer types: glioblastoma and thyroid cancer. We have also not found a *NISCH*-associated common gene network in GSEA analysis by sex. Male sex is associated with increased cancer risk and worse survival in many cancer types [88–91], and it could be that the differences in sample representation (in terms of grade, type etc.) in groups by sex were confounding our results. Nevertheless, this issue is worth further investigation, as a difference in metabolic phenotype of male and female NISCH KO mice was recently reported [15].

In contrast to the findings that exogenous expression of NISCH in breast cancer cells suppresses cell survival, *in vitro* studies from the early 2000s on the function of IRAS (then considered a human homologue of mouse nischarin) support the opposite claim. Overexpression of NISCH delayed apoptosis induced by a variety of stimuli [92, 93], partially through activation of the PI3 kinase pathway. Nischarin was also shown to bind insulin receptor substrate protein, activate ERK and promote survival [94]. Again, it is possible that NISCH signaling in terms of the cell survival is context-dependent, as is the case with some other genes, which can act as both tumor suppressors and proto-oncogenes (e.g. TGF-β, BRCA1, p16, p14, p53, etc.) [95]. Of importance, there are several FDA-approved antihypertensives with imidazoline ring that

are nischarin agonists: clonidine, moxonidine and rilmenidine [67]; as well as several endogenous ligands that are present in the brain tissue: agmatine, harmane, harmalan, and imidazoleacetic acid-ribotide [96]. Rilmenidine and rilmenidine-derived compounds were shown to induce apoptosis in breast [97] and leukemic cells *in vitro* [98]. Agmatine was found to have anti-proliferative [99] and anti-metastatic effects *in vitro* [100], and to suppress tumor growth of sarcomas and melanomas in mouse models [99]. Tizanidine hydrochloride reversed the proliferation, invasion, and migration of A549 cells caused by nischarin knockdown [5]. In our screen, rilmenidine decreased melanoma, pancreatic and colon cancer cell viability *in vitro*, and we confirmed that the most sensitive melanoma cell line A-375 is undergoing apoptosis in the presence of rilmenidine. Therefore, NISCH agonists present a great opportunity for testing as anti-cancer agents, at least in tumors in which NISCH is predicted to be a positive prognostic marker.

Taken together, our study highlights several novel findings with regards to the nischarin biology that are prompting further investigation: the nuclear localization in cancer, negative prognostic value in several cancer types (that questions the tumor suppressor role), and association with signaling pathways that regulate stemness in these cancer types. Our results lay a ground for further functional studies of the context-dependent nischarin role in cancer and investigation of the potential of NISCH agonization as a novel therapeutic approach in oncology.

## Supporting information

**S1 Fig. Nischarin expression in healthy tissue.**
(PDF)

**S2 Fig. NISCH protein expression summary from the HPA.**
(PDF)

**S3 Fig. NISCH isoform usage distribution across solid TCGA cancers according to the GEPIA2 website.**
(PDF)

**S4 Fig. *NISCH* prognostic value across different tumors.**
(PDF)

**S5 Fig. *NISCH* mRNA expression differences and Kaplan-Meier plots for left- and right-sided COAD.**
(PDF)

**S6 Fig. The effect of nischarin mRNA expression on overall survival of patients of the opposite sex.**
(PDF)

**S7 Fig. *NISCH* mutations in TCGA cancers.**
(PDF)

**S8 Fig. Copy-number alterations in TCGA cancers in which *NISCH* was identified as prognostic marker.**
(PDF)

**S1 Table. *NISCH* prognostic value across TCGA cancers with hazard ratio (HPA data) and adjusted by clinical factors (data from the TIMER2.0 website).**
(XLSX)

**S2 Table. GSEA of primary tumors in which *NISCH* was a significant prognostic marker—Hallmark.**
(XLSX)

**S3 Table. GSEA of primary tumors in which *NISCH* was a significant prognostic marker—KEGG.**
(XLSX)

**S4 Table. GSEA for GBM, SKCM and THCA primary melanoma samples divided by patients' sex—Hallmark.**
(XLSX)

**S5 Table. GSEA for GBM, SKCM and THCA primary melanoma samples divided by patients' sex—KEGG.**
(XLSX)

**S6 Table. GSEA for GBM, SKCM and THCA primary melanoma samples divided by patients' sex—Reactome.**
(XLSX)

## Acknowledgments

We thank Dr Miljana Tanić for the meaningful suggestions for the improvement of the study design. The results shown here were in part based upon data generated by the TCGA Research Network: https://www.cancer.gov/tcga.

## Author Contributions

**Conceptualization:** Jelena Grahovac.

**Data curation:** Marija Ostojić, Ana Đurić.

**Formal analysis:** Marija Ostojić, Kristina Živić.

**Funding acquisition:** Jelena Grahovac.

**Investigation:** Marija Ostojić.

**Supervision:** Jelena Grahovac.

**Visualization:** Marija Ostojić, Ana Đurić, Kristina Živić.

**Writing – original draft:** Marija Ostojić, Jelena Grahovac.

**Writing – review & editing:** Marija Ostojić, Jelena Grahovac.

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
