## [Decision Letter · Decision Letter 0]

8 Mar 2024

PONE-D-24-06040Analysis of the nischarin expression across human tumor types reveals its context-dependent role and a potential as a target for drug repurposing in oncologyPLOS ONE

Dear Dr. Grahovac,

Thank you for submitting your manuscript to PLOS ONE. After careful consideration, we feel that it has merit but does not fully meet PLOS ONE’s publication criteria as it currently stands. Therefore, we invite you to submit a revised version of the manuscript that addresses the points raised during the review process.

Please submit your revised manuscript by Apr 22 2024 11:59PM If you will need more time than this to complete your revisions, please reply to this message or contact the journal office at plosone@plos.org. Please include the following items when submitting your revised manuscript:A rebuttal letter that responds to each point raised by the academic editor and reviewer(s). You should upload this letter as a separate file labeled 'Response to Reviewers'.A marked-up copy of your manuscript that highlights changes made to the original version. You should upload this as a separate file labeled 'Revised Manuscript with Track Changes'.An unmarked version of your revised paper without tracked changes. You should upload this as a separate file labeled 'Manuscript'.

We look forward to receiving your revised manuscript.

Kind regards,

Chen Li, Ph.D.

Academic Editor

PLOS ONE

Journal Requirements:

   "This research was supported by the Science Fund of the Republic of Serbia, PROMIS Grant No. 6056979, REPANCAN to JG; by the Ministry of Education, Science and Technological Development of the Republic of Serbia Agreement No. 451-03-68/2022-14/200043 to all authors; and the European Union’s Horizon 2020 research and innovation programme under the Marie Skłodowska-Curie grant agreement No. 891135 to JG."

3. Please include a copy of Table 3,4 and 5 which you refer to in your text on page 31.

Reviewers' comments:

Reviewer's Responses to Questions

**Comments to the Author**

1. Is the manuscript technically sound, and do the data support the conclusions?

Reviewer #1: Yes

Reviewer #2: Yes

Reviewer #3: No

Reviewer #4: Partly

2. Has the statistical analysis been performed appropriately and rigorously? 

Reviewer #1: Yes

Reviewer #2: Yes

Reviewer #3: Yes

Reviewer #4: No

3. Have the authors made all data underlying the findings in their manuscript fully available?

Reviewer #1: Yes

Reviewer #2: Yes

Reviewer #3: Yes

Reviewer #4: Yes

4. Is the manuscript presented in an intelligible fashion and written in standard English?

Reviewer #1: Yes

Reviewer #2: Yes

Reviewer #3: Yes

Reviewer #4: Yes

5. Review Comments to the Author

Reviewer #1: Overall, the manuscript titled "Analysis of the nischarin expression across human tumor types reveals its context-dependent role and a potential as a target for drug repurposing in oncology" presents a comprehensive analysis of the expression and prognostic value of nischarin across various human tumor types. Below are comments to improve the manuscript:

1. Can you provide more insight into the cellular localization of nischarin in tumor tissues, particularly regarding its nuclear translocation? Are there any known molecular mechanisms driving this translocation, and how does it correlate with cancer-specific functions of nischarin?

2. Could you elaborate on the functional differences among nischarin isoforms and their potential contributions to cancer progression? Are there specific domains or motifs within nischarin isoforms that confer distinct functional properties or regulatory mechanisms?

3. Given the differential expression patterns of nischarin isoforms across tumor types, do you have any insights into the transcriptional or post-transcriptional regulatory mechanisms governing isoform expression in cancer cells?

4. Regarding the sex-specific differences in the prognostic value of nischarin, have you explored the underlying molecular mechanisms driving these disparities? Are there sex hormone-related pathways or signaling networks that intersect with nischarin activity in cancer cells?

5. In tumors where nischarin expression was associated with unfavorable prognosis, have you investigated the downstream signaling pathways or cellular processes regulated by nischarin? How do these findings inform potential therapeutic strategies targeting nischarin in aggressive cancers?

6. Regarding mutations in the NISCH gene, have you conducted functional assays to assess the impact of missense mutations on nischarin structure, localization, and function? Are there specific domains or interaction interfaces within nischarin that are more susceptible to mutational disruption?

7. In the context of nischarin agonist treatment, have you investigated the downstream signaling cascades or molecular pathways modulated by rilmenidine in cancer cells? Are there specific cellular processes or survival mechanisms targeted by nischarin agonists that contribute to their anti-cancer effects?

8. Given the observed differences in sensitivity to nischarin agonists among different cancer cell lines, have you explored the molecular determinants or biomarkers associated with responsiveness to nischarin-targeted therapies? How do these findings inform patient stratification and personalized treatment approaches in oncology?

Reviewer #2: The paper reports a study that explores nischarin's role in various cancers, showing it as context-dependent molecule rather than a universal tumor suppressor. Through analyzing public databases and in vitro experiments, it demonstrates variable prognostic values of nischarin across solid tumors and investigates the therapeutic potential of rilmenidine, a nischarin agonist. This research challenges existing paradigms by highlighting nischarin's varying impact on cancer progression and underscores the need for further study to fully understand its mechanisms and therapeutic implications. This study provides valuable insights into the understanding of nischarin’s role in cancer treatment. It is publishable if the following questions could be addressed.

1. This paper’s analysis showed that level of Nischarin in most of tumor cells are lower than healthy cells. However, the expression Nischarin is not a universal maker of good prognosis. These two conclusions seem contradictory. Could the author provide more explanation on it?

2. The survival of the colon was conducted by left and right side. Is it possible that the location may also matter for other types of cancers?

3. In the results section, authors declared that methylation was not an important mechanism for NISCH downregulation. But abstract says it could be. I suggest authors reorganize the language to avoid confusion.

4. In figure 7, the cell viability experiment shows that rilmenidine could inhibit cancer cells. Why there is no control group to show the ground read out?

Reviewer #3: Summary

Nischarin is identified as a tumor suppressor with a critical role in the initiation and progression of various cancers, including breast, ovarian, and lung cancers, displaying both tumor type-specific and sex-dependent prognostic values. Research revealed nischarin's expression is diminished in tumors due to gene deletions and promoter methylation, with its aberrant localization in the nuclei of tumor tissues suggesting a unique cancer-specific function. The study suggests nischarin's complex involvement in cancer progression and the potential for repurposing its agonists, like rilmenidine, to reduce cancer cell viability, especially in cases where nischarin is a positive prognostic marker.

Comments:

1. A potential disadvantage could be the study's reliance on public databases for the analysis of nischarin's prognostic value and associated signaling pathways, which might limit the scope to existing data and potentially overlook novel or underrepresented aspects of nischarin's role in various cancers.

2. While the study tests the effects of a nischarin agonist in vitro, these findings might not fully translate to in vivo contexts or clinical efficacy. The authors are suggested to validate their conclusions at least in relevant cancer cell lines.

3. The observation that nischarin can act as both a positive and negative prognostic marker across different cancer types and sexes suggests that its role in cancer progression is complex and might require more nuanced, context-specific studies to fully understand its therapeutic potential and limitations. More discussions might be added to this manuscript.

Reviewer #4: 1. The conclusions drawn about nischarin's context-dependent role in cancer progression and as a potential target for drug repurposing are intriguing. However, they seem to extend beyond what the data robustly support in some instances. The authors are encouraged to more closely align their conclusions with the evidence provided and discuss alternative interpretations of the data where applicable.

2. The manuscript would benefit from a more thorough description of the statistical analyses performed, including the tests used for each data type, justification for their selection, and how multiple testing was addressed. Specific concerns arise from the pan-cancer approach, where the potential for Type I errors increases. The inclusion of a section detailing these statistical considerations would greatly enhance the manuscript's rigor.

3. Adjusting for multiple comparison is needed and please re-perform all statistical analysis and provide the adjusted p value results.

4. HR is needed for survival analysis.

5. Multivariable analysis is needed for all the survival analysis to exclude the impact of clinical confounding factors such as gender, age, cancer stages.

6. Data resolution too low. NO label for IHC experiments and results.

7. While the manuscript is generally well-written, there are sections where the language could be simplified for clarity without sacrificing scientific accuracy. Technical jargon should be minimized or clearly defined upon first use to ensure the manuscript is accessible to a broad scientific audience, including those not specialized in oncology or molecular biology.

6. PLOS authors have the option to publish the peer review history of their article (what does this mean?). If published, this will include your full peer review and any attached files.

Reviewer #1: No

Reviewer #2: No

Reviewer #3: No

Reviewer #4: No

---

## [Author Response · Author response to Decision Letter 0]

19 Apr 2024

We thank the Academic Editor and the reviewers for the thorough analysis of our manuscript and the suggestions on how to improve it. 

For the Editorial office for the Funder statement: “The funders had no role in study design, data collection and analysis, decision to publish, or preparation of the manuscript”

We have now separated supplemental tables from one supplemental table with tabs, to separate Supplemental Table files, as requested.

For the reviewers, we have addressed the comments in point by point below:

Reviewer #1: 

1. Can you provide more insight into the cellular localization of nischarin in tumor tissues, particularly regarding its nuclear translocation? Are there any known molecular mechanisms driving this translocation, and how does it correlate with cancer-specific functions of nischarin?

In our previous study investigating nischarin expression in melanoma published last year (PMID: 37382661), our group reported that NISCH was also present in the cell nucleus in melanoma tissues, in addition to the cytoplasmic and membranous localization that was previously reported in breast cancer. Our observation of the nuclear localization was the first reported, and we have not elucidated yet which mechanisms are involved in translocation. In the current study we examined the subcellular distribution of NISCH across various tumor types in the HPA and found that only in breast and endometrial cancer NISCH localization was restricted to the cytoplasmic and membranous, but in the rest of the tumor types NISCH could also be found in the nucleus. This is a new discovery and there is no previous research dealing with this topic. Nuclear translocation could possibly account for cancer-specific nischarin role. We do plan to validate these observations in patient samples and in vitro models, but these are beyond the scope of the present manuscript, which is hypothesis generating. This is discussed in more detail in the section/lines 557-569 of discussion.

2. Could you elaborate on the functional differences among nischarin isoforms and their potential contributions to cancer progression? Are there specific domains or motifs within nischarin isoforms that confer distinct functional properties or regulatory mechanisms?

NISCH has 4 isoforms, but only isoform 1 is the full-length protein (1504aa, 167kDa). All the other isoforms lack important sequence parts of a full functioning NISCH protein. Discussion of the potential functional properties of shorter isoforms is now discussed in more detail in the section “Nischarin isoforms in healthy and cancer tissues” (lines 274-287). Namely, isoform 2 is missing amino acids 1-511, isoform 3 has a modified sequence in 511-583aa and is missing amino acids 584-1504, and isoform 4 is missing amino acids 516-1504 and differs from the isoform 1 in amino acids 512-515. Therefore, isoform 2 is missing an important part of N-terminus with PX domain, a domain of NISCH/IRAS that binds to phosphatidylinositol-3-phosphate in membranes, as well as leucine-rich region motifs important for protein-protein interactions. The PX domain together with the coiled-coil domain of NISCH is essential for its localization to endosomes, implying that NISCH isoforms 2, 3, and 4 may have different cellular localization. Additionally, positions 1–624aa of NISCH canonical sequence interact with PAK1; 416–624aa region interacts with LKB1 and LIMK; positions 464-562aa interact with the integrin α5 cytoplasmic tail; NISCH C-terminal domain interacts with IRS 1–4 and Rab14, and both the N- and C-terminus can interact with Rac1 (elaborated in Results section of the manuscript, under the Mutations headline). 

Given that in our results the most dominant NISCH transcript was the one coding for the full-length protein, and that in the majority of the tumors most of the examined transcripts were decreased, we hypothesize that in cancer there is no NISCH isoform switching and that isoform 1 is the dominant isoform both in healthy and cancer tissues. 

3. Given the differential expression patterns of nischarin isoforms across tumor types, do you have any insights into the transcriptional or post-transcriptional regulatory mechanisms governing isoform expression in cancer cells?

NISCH isoforms are produced by the alternative splicing (resulting in multiple transcript variants encoding different isoforms). To our knowledge, there are no studies investigating transcriptional or post-transcriptional regulatory mechanisms governing NISCH isoform expression in healthy cells or cancer cells. 

4. Regarding the sex-specific differences in the prognostic value of nischarin, have you explored the underlying molecular mechanisms driving these disparities? Are there sex hormone-related pathways or signaling networks that intersect with nischarin activity in cancer cells?

In our previous study (PMID: 37382661) in melanoma, in which gene set enrichment analysis (GSEA) suggested significant sex-related disparities in predicted association of NISCH with several signaling pathways, we found that sex-differences were associated with differences in tumor immune infiltration, and not with the hormone-related pathways per se. In the current manuscript we found the sex-differences in NISCH prognostic value in GBM, SKCM and THCA samples, but GSEA did not reveal the common thread of association. 

5. In tumors where nischarin expression was associated with unfavorable prognosis, have you investigated the downstream signaling pathways o cellular processes regulated by nischarin? How do these findings inform potential therapeutic strategies targeting nischarin in aggressive cancers?

We performed the gene set enrichment analysis (GSEA) to look for the differences in associated gene networks in tumor types where NISCH had negative versus positive prognostic value. GSEA showed that in tumors in which high nischarin expression was a negative prognostic marker, signaling pathways that regulate stemness (Wnt, Hedgehog and Notch) were enriched, discussed in lines 649-656. Since association of NISCH with Notch and Hedgehog signaling is a novel finding, further investigation and functional assays in each tumor type separately are needed to draw conclusions, as NISCH role seems to be context dependent. 

6. Regarding mutations in the NISCH gene, have you conducted functional assays to assess the impact of missense mutations on nischarin structure, localization, and function? Are there specific domains or interaction interfaces within nischarin that are more susceptible to mutational disruption?

This manuscript is hypothesis generating. We did not conduct functional assays to assess the impact of mutations, methylation or transcript variant expression. There is a possibility that mutations could affect some of the protein-protein interactions between NISCH and other signaling molecules and disrupt NISCH antitumor effects. However, in the examined datasets the incidence of mutations in the NISCH gene was so low that we concluded that it is not an important contributor to the loss of tumor suppressor function in certain cancer types. Mutations in the NISCH gene were present in most of the examined cancers and across the length of the gene, but with no specific clustering to be considered as a cause for negative prognostic value of NISCH in a subset of tumors.

7. In the context of nischarin agonist treatment, have you investigated the downstream signaling cascades or molecular pathways modulated by rilmenidine in cancer cells? Are there specific cellular processes or survival mechanisms targeted by nischarin agonists that contribute to their anti-cancer effects?

Our GSEA results showed that, regardless of the tumor type, decreased NISCH expression was associated with activation of metabolic pathways that allow increased tumor growth. These findings support our in vitro results that nischarin agonist, through an increase in the nischarin activity, has the opposite effect and decreases the viability of cancer cells. We have now added data to the Figure 7D, showing that rilmenidine treatment induces apoptosis in melanoma cells. We are currently performing validation studies for the anti-cancer potential of nischarin agonists in selected cancer types, but these are beyond the scope of the present manuscript and will be reported once the mechanistic studies are done. 

8. Given the observed differences in sensitivity to nischarin agonists among different cancer cell lines, have you explored the molecular determinants or biomarkers associated with responsiveness to nischarin-targeted therapies? How do these findings inform patient stratification and personalized treatment approaches in oncology?

In the present study we have screened melanoma, pancreatic and colon cancer cell lines and found that the melanoma lines were the most sensitive. Melanoma is highly metabolically active in vitro and this sensitivity will be further investigated in our future functional studies. In the current manuscript, we have now added the data that shows that the most sensitive line A-375 is indeed undergoing apoptosis in the presence of rilmenidine (Figure 7D). 

Reviewer #2: 

1. This paper's analysis showed that level of Nischarin in most of tumor cells are lower than healthy cells. However, the expression Nischarin is not a universal maker of good prognosis. These two conclusions seem contradictory. Could the author provide more explanation on it?

Nischarin is known for its role in the control of cell migration in cancer and our GSEA results confirmed that, regardless of the tumor type, decreased NISCH expression was associated with activation of metabolic pathways that allow increased tumor growth. This is in line with our findings that nischarin levels are lower in most cancers compared to healthy tissues, since these are critical events for tumor growth, motility and invasion. However, the full contribution of nischarin in the progression of different cancer types is still unknown, since NISCH was mostly investigated in breast cancer. NISCH complex association with distinct metabolic and other signaling pathways further complicates elucidation of NISCH role (also see reply to R4 Q1).

It is not uncommon for a protein to have a context dependent tumor associated role (e.g. PMID: 29136504, PMID: 34354941). Our findings of NISCH association with signaling pathways that regulate stemness (Wnt, Hedgehog and Notch) in tumors in which NISCH was a negative prognostic marker, imply that NISCH role is context-dependent, but further investigation and functional assays in each tumor type separately are needed to draw conclusions. We have now discussed this aspect in more detail in the Discussion section (lines 640-659).

2. The survival of the colon was conducted by left and right side. Is it possible that the location may also matter for other types of cancers?

Left and right colon are anatomically distinct locations with distinct incidence of colon cancer in men and women and hypothesized biological differences in tumor development. For this reason, left and right localization is reported in most datasets. We have used this data to analyze possible differences. 

While there is evidence that the anatomic location can be a prognostic factor for survival in several cancer types, these differences stem from the availability for earlier diagnosis and surgical removal (e.g. in pancreas head vs tail (PMID: 18982154). While there is some information on the location for certain cancer types (e.g. in melanoma trunk vs. head vs. extremities), there was not enough data to analyse the association of NISCH expression with anatomical localization in other cancer types. 

3. In the results section, authors declared that methylation was not an important mechanism for NISCH downregulation. But abstract says it could be. I suggest authors reorganize the language to avoid confusion. 

In the results section, we stated that out of the 13 TCGA cancers analyzed (originally 14, but ovarian cancer was later excluded from the analysis due to lack of data) for the presence of NISCH promoter methylation, seven cancer types showed significant increase in methylation levels in tumor samples with lower NISCH expression compared to the samples with higher NISCH expression. This indicates that NISCH promoter methylation is frequent in cancer and can affect NISCH expression. 

We apologize if this was not communicated effectively and have now changed the text to clarify this (lines 441-443).

4. In figure 7, the cell viability experiment shows that rilmenidine could inhibit cancer cells. Why there is no control group to show the ground read out?

In the standard MTT assay protocol, positive controls are the cells not treated with the investigated drug. The concentration of 0 corresponds to the control and the results are expressed relative to the 100% viability in control. The IC50 values shown in figure 7B are concentrations of the investigated compound that caused 50% decrease in the MTT reduction in treated cell population compared to a non-treated control, read out from the graph in 7C. We have now clarified this in the methods section (lines 194-195). 

Reviewer #3: 

1. A potential disadvantage could be the study's reliance on public databases for the analysis of nischarin's prognostic value and associated signaling pathways, which might limit the scope to existing data and potentially overlook novel or underrepresented aspects of nischarin's role in various cancers.

Although the use of publicly available data has some limitations, following the FAIR data and Open Science principles in modern oncology research is of essential importance. Beyond proper data collection, annotation, and archival, the goal of Open Science is that the data should be available, re-discovered and re-used by investigators, alone or in combination with newly generated data. In this way, the conclusions we obtain based on the data generated by different research groups have greater value. We agree that relying on publicly available information puts certain restraints on availability of the data for each cancer type and lowers adherence on standards that hold when one performs analysis of the tissue that is available in house. Validating these findings in prospective studies for each cancer type is of great interest for our group. Besides its limitations, our manuscript is a comprehensive pan-cancer analysis that aims at generating novel hypotheses on nischarin role in cancer and we unraveled a plethora of intriguing cues that are worth investigating in functional assays. 

2. While the study tests the effects of a nischarin agonist in vitro, these findings might not fully translate to in vivo contexts or clinical efficacy. The authors are suggested to validate their conclusions at least in relevant cancer cell lines.

In the present study we have screened representative melanoma, pancreatic and colon cancer cell lines and found that the melanoma cell line A-375 was the most sensitive. Melanoma is highly metabolically active in vitro and this sensitivity will be further investigated in our future functional studies. In the current manuscript, we have now added the data that shows that the most sensitive line A-375 is undergoing apoptosis in the presence of rilmenidine. This is now shown in Figure 7D. Mechanistic validation in in vitro and in vivo setting in this and other cancer types is part of our future efforts, and is beyond the scope of the current manuscript.

3. The observation that nischarin can act as both a positive and negative prognostic marker across different cancer types and sexes suggests that its role in cancer progression is complex and might require more nuanced, context-specific studies to fully understand its therapeutic potential and limitations. More discussions might be added to this manuscript.

We admit that the findings on the negative prognostic value of NISCH in several cancer types are the most intriguing and have now discussed them in more detail in the discussion section (lines 642-648). 

Reviewer #4: 

1. The conclusions drawn about nischarin's context-dependent role in cancer progression and as a potential target for drug repurposing are intriguing. However, they seem to extend beyond what the data robustly support in some instances.

---

## [Decision Letter · Decision Letter 1]

8 May 2024

Analysis of the nischarin expression across human tumor types reveals its context-dependent role and a potential as a target for drug repurposing in oncology

PONE-D-24-06040R1

Dear Dr. Jelena Grahovac,

We’re pleased to inform you that your manuscript has been judged scientifically suitable for publication and will be formally accepted for publication once it meets all outstanding technical requirements.

Kind regards,

Chen Li, Ph.D.

Academic Editor

PLOS ONE

Additional Editor Comments (optional):

Reviewers' comments:

Reviewer's Responses to Questions

**Comments to the Author**

1. If the authors have adequately addressed your comments raised in a previous round of review and you feel that this manuscript is now acceptable for publication, you may indicate that here to bypass the “Comments to the Author” section, enter your conflict of interest statement in the “Confidential to Editor” section, and submit your "Accept" recommendation.

Reviewer #2: All comments have been addressed

Reviewer #3: All comments have been addressed

2. Is the manuscript technically sound, and do the data support the conclusions?

Reviewer #2: Yes

Reviewer #3: Yes

3. Has the statistical analysis been performed appropriately and rigorously? 

Reviewer #2: Yes

Reviewer #3: Yes

4. Have the authors made all data underlying the findings in their manuscript fully available?

Reviewer #2: Yes

Reviewer #3: Yes

5. Is the manuscript presented in an intelligible fashion and written in standard English?

Reviewer #2: Yes

Reviewer #3: Yes

6. Review Comments to the Author

Reviewer #2: All my questions have been addressed. The paper reports a study that explores nischarin's role in various cancers, showing it as context-dependent molecule rather than a universal tumor suppressor. It is a valuble paper to be published

Reviewer #3: This revision demonstrates a significant improvement; the authors have addressed all of my previous comments and concerns. I don’t have any further questions.

7. PLOS authors have the option to publish the peer review history of their article (what does this mean?). If published, this will include your full peer review and any attached files.

Reviewer #2: No

Reviewer #3: No

---

## [Editor Report · Acceptance letter]

13 May 2024

PONE-D-24-06040R1 

PLOS ONE

Dear Dr. Grahovac, 

I'm pleased to inform you that your manuscript has been deemed suitable for publication in PLOS ONE. Congratulations! Your manuscript is now being handed over to our production team.

Kind regards, 

on behalf of

Dr. Chen Li 

Academic Editor

PLOS ONE